# Who Evaluates the Evaluations? Objectively Scoring Text-to-Image Prompt Coherence Metrics with T2IScoreScore (TS2)

**Michael Saxon**[⊗◎]   **Fatima Jahara**[⊗□△]   **Mahsa Khoshnoodi**[⊗△]
**Yujie Lu**[◎]   **Aditya Sharma**[◎]   **William Yang Wang**[◎]

[◎]University of California, Santa Barbara   [□]Rutgers University
[△]Fatima Al-Fihri Predoctoral Fellowship   [⊗]*Equal contribution*

*Contact:* **saxon@ucsb.edu**

**T2IScoreScore.github.io**

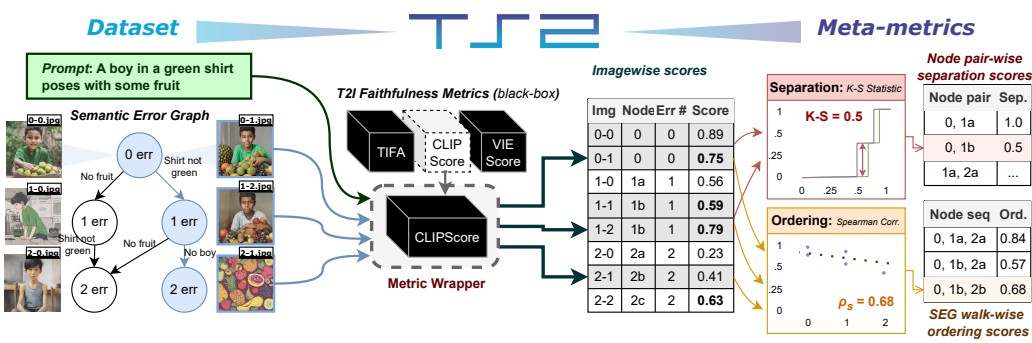

Figure 1: Overview of **T2IScoreScore**. T2I evaluation metrics are scored based on their ability to correctly organize images in a *semantic error graph* (SEG) relative to their generating prompt, checking ordering (Spearman's $\rho$) and separation of nodes (Kolmogorov–Smirnov statistic).

## Abstract

With advances in the quality of text-to-image (T2I) models has come interest in benchmarking their *prompt faithfulness*—the semantic coherence of generated images to the prompts they were conditioned on. A variety of T2I faithfulness metrics have been proposed, leveraging advances in cross-modal embeddings and vision-language models (VLMs). However, these metrics are not rigorously compared and benchmarked, instead presented with correlation to human Likert scores over a set of easy-to-discriminate images against seemingly weak baselines.

We introduce **T2IScoreScore**, a curated set of *semantic error graphs* containing a prompt and a set of increasingly erroneous images. These allow us to rigorously judge whether a given prompt faithfulness metric can correctly order images with respect to their objective error count and significantly discriminate between different error nodes, using meta-metric scores derived from established statistical tests. Surprisingly, we find that the state-of-the-art VLM-based metrics (e.g., TIFA, DSG, LLMScore, VIEScore) we tested fail to significantly outperform simple (and supposedly worse) feature-based metrics like CLIPScore, particularly on a hard subset of naturally-occurring T2I model errors. **TS2** will enable the development of better T2I prompt faithfulness metrics through more rigorous comparison of their conformity to expected orderings and separations under objective criteria.

38th Conference on Neural Information Processing Systems (NeurIPS 2024).

# 1  Introduction

Text-to-image (T2I) models are improving at a breakneck pace in terms of quality, fidelity, and coherence of generated images to their conditioning prompts [1–4]. Despite this, persistent challenges in achieving image-prompt faithfulness [5, 6] remain—particularly in freely available models that don't sit behind proprietary APIs. Indeed, many techniques to improve T2I models have been proposed of late, aiming to reduce hallucination [7, 8], duplication [9], composition errors [8, 10], and missing objects [11, 12]. However, there is no consensus on how to best compare these many models and methods, so it is hard to objectively track T2I progress [13, 14].

Recent work has proposed a litany of automated *image-prompt coherence metrics* which rate the *faithfulness* of generated images: the degree to which a they satisfy the implicit requirements set forth in the generating prompt [15–18]. These proposed metrics vary considerably in design; as rating how well an image matches to its prompt is a nontrivial multimodal challenge [14, 19, 20].

This variety itself presents a *meta-evaluation problem*: there is no **consensus on how these faithfulness metrics ought to be compared**, and consequently each new metric is validated on its own ad-hoc test set against prior baselines. Typically these *self-evaluations* consist of a set of prompt-image pairs with accompanying human annotations (usually simple Likert scores [16, 19]), and metrics are judged on their correlation to these human judgements [20].

Such self-evaluation is not ideal; authors may unwittingly tilt the scales by using evaluation examples which cater to the particular strengths of their proposed method, and variance of metric performance between different evaluation sets (containing different images and prompt semantics [21, 22]) is high [23]. Additionally, relying on correlation to human judgements of small sets of images across different prompts is highly subjective [24, 25] and prone to including judgements of quality and style that are orthogonal to prompt coherence. We need a consistent and objective meta-evaluation.

To this end we propose **T2IScoreScore** (**TS2**), a *benchmark and set of meta-metrics* for evaluating T2I faithfulness metrics. While it contains a similar number of images to previously proposed coherence metric evaluation sets, it contains fewer prompts. This high image-to-prompt ratio allows us to organize the images along *semantic error graphs*, or SEGs (fig. 1), where each edge corresponds to a specific error with respect to the prompt that a child image set possesses but its parent images do not. These semantic error graphs permit objective scoring of a metric by answering:

1. Can a metric correctly **order** increasingly wrong images against their generating prompt?
2. Can a metric reliably **separate** sets of images that differ by a specific semantic error?
3. Does the metric **confidently** separate the image sets within its dynamic range?

We adapt existing statistical tests [26, 27] to the SEG setting to answer these questions for a broad set of T2I faithfulness metrics. We find some surprising results: despite their inferior performance in correlating to human preferences against complicated vision-language model (VLM)-based metrics [6, 16–18], simple embedding-correlation methods like CLIPScore [15] are actually quite performant on our meta-metrics, and **Pareto-optimal** with respect to compute cost (§5). In summary, we:

- Formalize the task of objectively assessing T2I prompt coherence metrics by their ability to correctly order and separate image populations within semantic error graphs (SEGs). (§2)
- Present **T2IScoreScore** (**TS2**), our evaluation for this task: a carefully-curated benchmark dataset of SEGs each containing between 4 and 76 images, permitting 93,000 total pairwise image comparisons and meta-metrics for ordering and separation in SEGs. (§2, §3)
- Evaluate a broad and representative set of T2I faithfulness benchmarks using **TS2**, demonstrate that it identifies novel failure cases, and motivate future work on improved metrics. (§4 §5, §6)

## 1.1  Related Work

Most evaluations in the T2I space test the quality of generating models based on *a fixed faithfulness metric's scores over a fixed benchmark set of prompts*. Often these reference prompt sets are designed for testing a single specific capability. DrawBench [4], T2I-CompBench [28], and ABC-6k [8] focus on attributes like compositionality, cardinality, and spatial relations, in strictly text-guided image generation, while ImagenHub [14] tests them in a broader set of settings like image editing and subject-driven synthesis. Other evaluation dimensions such as multilinguality [29, 30] and stereotype bias [31] have also been explored. These prompts are usually sourced from some combination of

| Dataset | # Images | | Img. per Prompt | | # Img Comparisons | | Ad-hoc |
| | Total | T2I-Gen. | Avg | Per equiv. pref | Total | T2I Errors | |
|---|---|---|---|---|---|---|---|
| *Benchmarks for captioning models.* | | | | | | | |
| Flickr8k [32] | 8k | 0 | 0.2 | 0.2 | 8k | 0 | – |
| Flickr30k [33] | 31k | 0 | 0.2 | 0.2 | 31k | 0 | – |
| MSCOCO Captions [34] | 330k | 0 | 0.22 | 0.22 | 330k | 0 | – |
| *Benchmarks for image retrieval/matching models. (Could be used for T2I metric evaluation)* | | | | | | | |
| SeeTRUE [38] | 31k | 0 | 1 | 1 | 31k | 0 | ✓ |
| Pick-a-Pic [37] | 500k | 500k | 2[1] | 1 | 7M | 0 | ✓ |
| *Benchmarks for T2I faithfulness metrics.* | | | | | | | |
| TIFA v1.0 [16] | 800 | 800 | 5 | 1 | 4k | 0 | ✓ |
| DSG-1k [16] | 1k | 1k | 1 | 1 | 1k | 0 | ✓ |
| **T2IScoreScore** | *2.8k* | *2690* | **17** | **3.4** | *93k* | **3.1k** | – |

Table 1: Comparison of benchmark datasets that can be used to evaluate T2I faithfulness. *Per equiv pref* means the average number of images for each prompt that are assigned the same preference or correctness score. **Bold** numbers are best overall, *italic* are best of the T2I metric benchmarks.

existing natural image captioning resources [32–34] and sets of in-the-wild conditioning prompts produced by real users [35, 36]. These benchmarks assess image quality either by direct human analysis or automated metrics [14] including those we analyze in this work. Often the goal of these benchmarks is to analyze how well a model generates images that comport with human preferences [14, 37], and directly elicit opinions from users through a web interface for this purpose.

To meta-evaluate faithfulness metrics *a benchmark containing a fixed set of images and prompts is required*. However, all existing benchmarks for this purpose suffer from two key limitations:

1. Evaluating on noisy *human preference* rather than *explicitly labeled objective differences*
2. *Low image-to-prompt ratio* limiting evaluation of discriminatory power over similar images

Captioning benchmarks [32–34] are poor candidates for faithfulness metric evaluation as single images are paired with multiple prompt candidates rather than vice versa. Image matching and entailment benchmarks such as SeeTRUE [38] and Pick-a-Pic [37] are also limited by a low ratio.

The few extant deliberately-designed faithfulness evaluation sets are limited by both factors. TIFA v1.0 [16] and DSG-1k [6] were proposed *ad-hoc* to demonstrate the utility of their accompanying metrics by relating the scores assigned by the metric to human preferences. These are done over small sets of images (800 & 1000 respectively) with slightly more images-per-prompt (5 & 1).

The two limitations of these prior evals are linked. A reliance on human preference correlations is a natural consequence of having few and poorly organized images to compare to each prompt. A lack of meta-metrics designed for evaluating structured aspects other than human preference correlation means limited utility in collecting larger, structured sets of images for each prompt. By providing both *meta-metrics* **and** *a structured eval set* **T2IScoreScore** overcomes these limitations (table 1).

## 2 T2IScoreScore meta-metrics

We introduce three measures of ordering and separation by a given metric within semantic error graphs (SEGs). We define SEG $S$ as prompt $P$ and a directed acyclic graph of nodes $n_i$ containing one or more images $I_j$ sharing the same errors wrt. the prompt. We label each node by its error count and type (eg, [0, 1a, 1b] has 1 node with 0 errors, and 2 nodes with 1 error each of different type).

A good prompt coherence metric will correctly rank images along each walk of increasing error counts within a SEG, and separate the scores assigned to images in successive nodes. Our metrics assess this by evaluating each walk separately. For ease of notation, we refer to each SEG as a set of walks $W \in S$ over nodes of increasing error count (eg, (0, 1a, 2a), (0, 1a, 2b), etc), where each walk is the in-order set of all (image, prompt, num. error) triples $(I, P, N) \in W$. For example, the first walk in fig. 1 is [(0-0.jpg, $P$, 0), (0-1.jpg, $P$, 0), (1-0.jpg, $P$, 1),(2-0.jpg, $P$, 2), ...].

We introduce measures of metric $m$ for SEG $S$: $\text{rank}_m(S)$, $\text{sep}_m(S)$ & $\text{delta}_m(S)$, assessed over every walk $W \in S$ in all SEGs in the **TS2** dataset to score a metric. They're defined as:

---

[1] Although this benchmark has 14 pairs of images per prompt, each pair is separately annotated. With no way to compare between pairs, the effective number of images per prompt is 2, with many repeated prompts.

## 2.1 Ordering score over walks: $\texttt{rank}_m$

We use Spearman's rank correlation coefficient $\rho$ [26] between image-level error count and metric-assigned score over every walk on a SEG to assess how a metric's faithfulness score aligns to our objective structure error counts. Spearman's $\rho$ is the PCC of the rank order of variables $X, Y$:

$$\rho(X, Y) = \frac{\text{cov}(R(X), R(Y))}{\sigma_{R(X)}\sigma_{R(Y)}}; \quad R(X) = \big\{ \sum_{x_i \in X} \mathbb{1}(x_i < x) \mid x_i \in X \big\} \tag{1}$$

Thus, in our case the SEG-level rank order score $\texttt{rank}_m(S)$ for scoring model $m$ is defined as:

$$\texttt{rank}_m(S) = \frac{1}{|S|} \sum_{W \in S} r_s(\{m(I, P) | (I, P, N) \in W\}, \{N | (I, P, N) \in W\}) \tag{2}$$

One limitation of Spearman's $\rho$ for characterizing scores is that it is undefined if one set $R(U)$ exclusively contains identical elements, as $\sigma_{R(U)} = 0$. For tractability in these scenarios we define $\rho(\cdot, R(U)) := 0$. If a metric assigns identical scores to all examples across different error levels, it presents no discernible relationship between error severity and score for that image set.

## 2.2 Statistical separation of error populations score: $\texttt{sep}_m$

We assess the two-sample Kolmogorov–Smirnov statistic [27] pairwise between the populations of metric $m$'s scores assigned to each sample between two error nodes $n_i$ and $n_j$ as populations. The Kolmogorov–Smirnov statistic is a non-parametric measure of the separation between two distributions [39, 40], defined as the maximum vertical difference between their empirical cumulative distribution functions $F_X(s)$:

$$D_{KS}(X, Y) = \sup_{x \in R_m} |F_X(x) - F_Y(x)| \tag{3}$$

Where $F_X(x)$ is proportion of samples in population $X$ for which the metric-assigned score $m(i) \leq x$, (see fig. 8 for a visual depiction). We compute $D_{KS}$ for every pair of adjacent error nodes in each tree walk $W$[2], and report the average over all of these as the SEG separation score $\texttt{sep}_m(S)$:

$$\texttt{sep}_m(S) = \frac{1}{|S|} \sum_{n_i \in S} D_{KS}(\{m(P, I) | (P, I) \in n_i\}, \{m(P, I) | (P, I) \in n_{i+1}\}) \tag{4}$$

## 2.3 Separation of nodes within dynamic range: $\texttt{delta}_m$

While $\texttt{sep}_m$ nonparametrically estimates whether pairs of nodes are drawn from different distributions (and thereby distinguished), it provides no information about the distance by which they are separated within the metric's dynamic range. This measure gives an alternative look at separation between nodes: the more separation a metric provides between nodes, the less severe slight variations in assigned score will be to ignoring errors in generated images. We assess it as:

$$\texttt{delta}_m(S) = \frac{1}{|S|\sigma_{m(\forall S)}} \sum_{N_i \in W} \text{avg}(\{m(P, I) | (P, I) \in N_i\}) - \text{avg}(\{m(P, I) | (P, I) \in N_{i+1}\}) \tag{5}$$

Where $\sigma_{m(\forall S)}$ is the standard deviation of scores from metric $m$ on all images in all SEGs in **TS2**. Our score is the average distance between the mean metric score of all adjacent nodes in all SEGs, rescaled by the standard deviation of the score to normalize against the metric's dynamic range.

## 3 The **T2IScoreScore** Dataset

We now turn to describing the **TS2** dataset collection process. We use three different semantic error graph collection procedures to produce a diverse set of SEGs. Each contains one prompt and between 4 and 76 images assigned to error nodes. Each node usually contains more than one image, though for simplicity in presentation we only show one image assigned to each node in this section.

---

[2]We use individual nodes $n_i \in S$ containing pairs of prompt, image $(P, I) \in n_i$ to make this equation easier to read.

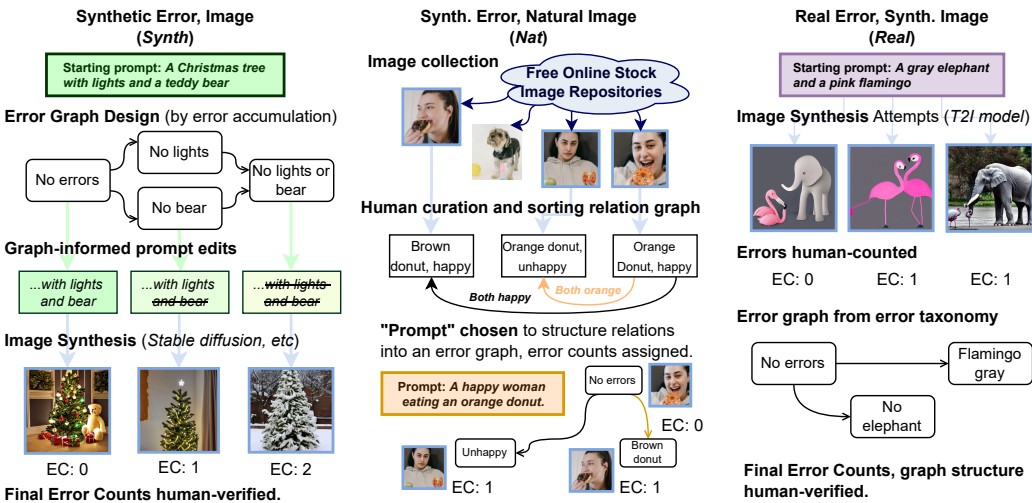

Figure 2: The three *semantic error graph* production procedures. **Synth.** (images generated from multiple prompts written to populate a SEG), **Nat.** (natural images populate a SEG), and **Real** (real errors from image generation attempts from one prompt populate a SEG).

## 3.1 Dataset Collection Procedure

Figure 2 depicts the three procedures by which we produce and populate SEGs with images: **synth**etic images from a synthetic graph (*Synth*), graph from **nat**ural images (*Nat*), and graph from **real** errors of synthetic images (*Real*), differentiated by prompt source, image source, and the order of production.

**Synth.** Synthetic SEGs are produced "graph first." From an initial prompt we list all entities and properties it contains, then ablate them to produce an error graph. We then manually write prompts describing each node, generate their images, and manually check image-node faithfulness. For example, in the left panel of fig. 2, the initial prompt "a Christmas tree with lights and a teddy bear" is converted to error prompts such as "a Christmas tree with lights."

**Nat.** The natural error trees exclusively contain real images sourced from the free stock image repository Pexels. We generate SEGs in "image, graph, prompt" order. We source sets of natural images that share objects and models. We organize them by relation graphs describing how the images differ by objects, actions, attributes, and composition. We then select a head node in this relation graph and write a "prompt" describing this head node (eg., fig. 2 center panel). We produced SEGs of natural images to assess whether distributional differences between synthetic images and real images might lead to measurable impacts on performance for the faithfulness metrics that typically use base models pretrained exclusively on natural images [41].

**Real.** The real error, synthetic image SEGs are produced following a "prompt, image, graph" order. From a seed prompt (both manually written and sourced from COCO or PartiPrompts) we simply generate a large set of images using a T2I model, then annotate the errors in each generated image. These error-labeled images are then organized into a final error graph for the SEG. This procedure is documented in the right panel of fig. 2.

## 3.2 Dataset structure, size, and validity

Each of the 165 SEGs in **TS2** is manually checked by three human annotators. The head node of each SEG contains at least one image that has been assessed by the annotators to contain no errors of verbal information (eg, entity in the image isn't performing the described action), compositionality (eg, object described as "on top" but is beneath object), missing objects, or incorrect object attributes.

Each edge on the SEG represents an error of one of the aforementioned types. Each node is labeled with the number of edges along its shortest path back to node 0, representing its error count (fig. 1, fig. 8). Each node contains at least one image which is erroneous according to the described errors.

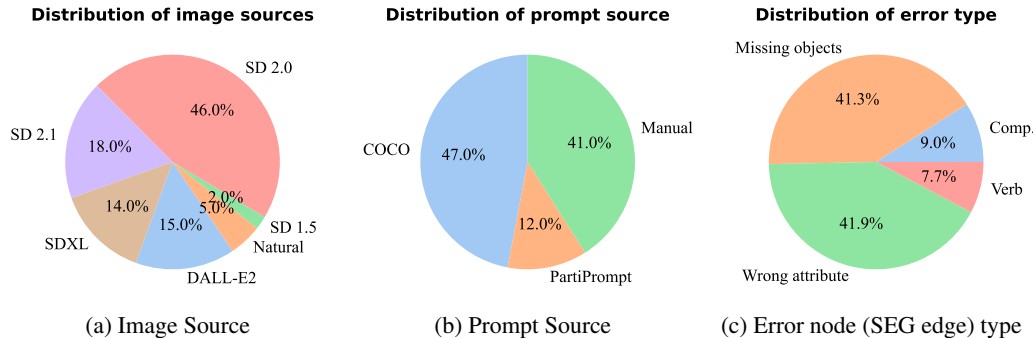

|              |              |              |
|:------------:|:------------:|:------------:|
| (a) Image Source | (b) Prompt Source | (c) Error node (SEG edge) type |

Figure 3: Overview of the distribution of sample types in **TS2**: (a) Where source images came from: 5% of images in the benchmark are real photographs from Pexels, while the remainder were generated by Stable Diffusion (SD) or DALL-E variants. (b) Source of the eliciting prompt; either existing resources or us (Manual). (c) Distribution of error types edges in all SEGs.

The synthetic images are generated with several T2I models—including DALL-E 2 [2], Stable Diffusion 1.5 [3], 2.0, 2.1, and SDXL [42]. We use MS-COCO [43], PartiPrompt [35], and manually written prompts in head nodes for the Synth and Real subsets. Image sources, prompt sources, and error type statustics are documented in fig. 3.

The total number of images and prompts is roughly in line with previous benchmarks that have been used to verify methods such as TIFA [16] and DSG [6] in their own papers, and permits significantly more image comparisons-per-prompt than any prior benchmark (Table 1), which we submit is the primary type of comparison required to verify that prompt-faithfulness assessments work.

## 4    Experiments

Using **TS2** we evaluated three classes of T2I evaluation metrics: *embedding-correlation* (comparing embeddings of prompt and image), *QG/A* (using VQA to check if requirement questions generated from the prompt are satisfied) and *caption-based* (comparing captions extracted from the generated images to the prompt). We evaluate these metrics with multiple backend VLMs (§A.2).

For each metric, we score every image against its SEG's prompt. We report the results of our Ordering and Separation metrics across all SEGs, as well as for our the Synth, Nat, and Real SEG subsets.

### 4.1    Embedding-correlation Metrics

**CLIPScore** [15] is a popular prompt faithfulness metric based on simple text-image similarity. CLIPScore is computed as the cosine similarity of the $L_2$-normalized CLIP-assessed [41] image and text embeddings. Equations and details in §A.1.

**ALIGNScore** (not to be confused with the text-only *AlignScore* [44]) is a variant embedding-based similarity score we produced using the ALIGN [45] embedding model rather than CLIP to embed the prompt and image. Other than model it is equivalent to CLIPScore.

### 4.2    Question Generation & Answering (QG/A) Metrics

*Question Generation & Answering Metrics* use an LM $\mathcal{M}_{QG}$ to produce a set of requirement question/answer pairs $(q, a) \in Q$ from prompt $p$, and then use a vision-language model $\mathcal{M}_{VL}$ to check each requirements against the image, reporting satisfaction rate as the image's faithfulness score. QG/A metrics vary by how questions are generated and relate to each other. Equations in §A.1.

**TIFA** [16] prompts an LM (GPT-3) to generate a set of multiple choice and yes-no questions and their expected answers relative to the prompt. Then a vision language model $\mathcal{M}_{VL}$ produces "free-form" answers to each question, which are converted into multiple choice answers $a'$ using an SBERT model. The TIFA score for a given image is then the rate of correct answers.

| | | CLIPScore | ALIGNScore | mPLUG | LLaVA 1.5 | LLaVA 1.5 (alt) | InstructBLIP | BLIP1 | Fuyu | GPT-4-V | mPLUG | LLaVA 1.5 | LLaVA 1.5 (alt) | InstructBLIP | BLIP1 | Fuyu | GPT-4-V | LLMScore EC | LLMScore Over | VIEScore |
|---|---|---|---|---|---|---|---|---|---|---|---|---|---|---|---|---|---|---|---|---|
| | | Emb-based | | TIFA | | | | | | | DSG | | | | | | | Caption-based | | |
| rank$_m$ | Avg | 71.4 | 73.9 | 71.0 | 74.5 | 74.4 | 76.5 | 73.8 | 38.7 | 77.9 | 70.4 | 76.2 | 75.0 | 79.0 | 76.6 | 29.5 | **79.6** | 48.8 | 57.7 | 37.8 |
| | Synth | 75.0 | 77.6 | 72.6 | 79.2 | 79.2 | 80.2 | 78.8 | 44.5 | 83.6 | 74.6 | 80.1 | 81.6 | **85.1** | 81.6 | 35.4 | 82.6 | 50.2 | 61.6 | 42.5 |
| | Nat | 58.0 | 70.2 | 66.9 | 62.8 | 64.0 | 65.1 | 62.2 | 23.5 | 61.6 | 65.3 | 65.9 | 68.8 | 70.7 | 71.6 | 20.5 | **73.7** | 36.2 | 44.4 | 22.4 |
| | Real | 69.3 | 62.6 | 68.2 | 66.7 | 64.5 | 71.6 | 64.0 | 29.7 | 69.5 | 58.4 | 70.0 | 54.2 | 62.0 | 61.2 | 14.2 | **73.0** | 54.4 | 54.1 | 33.2 |
| sep$_m$ | Avg | 90.7 | **92.8** | 80.6 | 82.5 | 81.9 | 85.0 | 81.8 | 67.2 | 83.2 | 78.4 | 83.1 | 80.3 | 84.2 | 80.8 | 63.6 | 84.2 | 73.6 | 73.5 | 51.8 |
| | Synth | 90.5 | **94.1** | 80.6 | 85.5 | 85.2 | 86.7 | 84.1 | 67.3 | 86.2 | 80.9 | 85.7 | 83.9 | **87.8** | 84.9 | 65.8 | 86.6 | 71.1 | 72.8 | 53.7 |
| | Nat | 91.5 | **92.6** | 84.2 | 75.1 | 75.6 | 82.8 | 77.9 | 75.7 | 80.5 | 71.2 | 80.9 | 76.7 | 81.8 | 73.3 | 68.6 | 81.7 | 80.5 | 76.7 | 44.5 |
| | Real | **90.3** | 87.9 | 77.4 | 76.8 | 74.4 | 80.5 | 76.4 | 59.3 | 73.7 | 75.1 | 74.5 | 69.4 | 71.9 | 70.8 | 50.3 | 76.6 | 77.3 | 73.6 | 50.7 |
| delta$_m$ | Avg | 89.7 | 95.6 | 92.4 | 97.5 | 97.1 | 99.8 | 94.0 | 43.9 | 92.3 | 94.2 | 104.7 | 102.0 | **110.6** | 103.7 | 37.7 | 110.2 | 66.9 | 56.3 | 45.9 |
| | Synth | 89.9 | 95.6 | 92.5 | 97.6 | 97.3 | 99.9 | 93.9 | 44.6 | 92.5 | 94.4 | 104.7 | 102.1 | **110.7** | 103.6 | 37.8 | 110.1 | 67.2 | 56.7 | 46.7 |
| | Nat | 92.5 | 98.6 | 94.4 | 98.7 | 98.8 | 101.1 | 95.5 | 46.1 | 93.9 | 96.0 | 105.5 | 103.0 | **111.6** | 104.6 | 39.3 | 111.5 | 66.2 | 56.4 | 46.6 |
| | Real | 95.1 | 100.8 | 96.0 | 100.5 | 100.4 | 102.6 | 96.9 | 46.9 | 95.9 | 97.3 | 106.8 | 104.2 | **113.0** | 105.9 | 39.4 | **113.0** | 65.0 | 56.0 | 46.3 |
| FLOP/run | | **604M** | 688M | 224T | 224T | 224T | 224T | 224T | 224T | 1.66P | 140T | 140T | 140T | 140T | 140T | 140T | 860T | 7.01T | 7.01T | 2.6T |

Table 2: Spearman ordering score rank$_m$, and Kolmogorov–Smirnov separation score sep$_m$, average dynamic range delta delta$_m$, (all reported as % for readability) and estimated FLOPs to score an image for each model. Best **bold**, within 2% of best underlined, top four colored by type (emb-based, TIFA, DSG, caption-based). See §A.3 for information on how we estimate compute costs.

**DSG** (*Davidsonian scene graph*) [6] shares the QA structure of TIFA, but generates a set of requirement questions which are non-overlapping, have exclusively yes/no answers, and sit on a directed acyclic graph such that a question is only satisfied if it and all its parent questions are answered yes.

**Backend VLMs.** All QG/A metrics rely on the use of a generative vision-language model (VLM) either for performing question answering ($\mathcal{M}_{VL}$ in §4.2) or captioning ($\mathcal{M}_C$ in §4.3). Thanks to the simple decomposable framework of the QG/A methodologies, we were able to efficiently test the performance of both TIFA and DSG using several VLMs as visual question-answering backends MVL$\mathcal{M}_{VL}$. We used mPLUG, LLaVA, BLIP, InstructBLIP, Fuyu, and GPT-4V as VLM backends for the QG/A metrics. Details and reference for each VLM is provided in appendix §A.2.

### 4.3 Caption-comparison Metrics

**LLMScore** [17] captures the fine-grained similarity between the image and text with rationales by leveraging the visual details understanding capability from vision experts and the reasoning capability of LLMs. The visual information is parsed in hierarchical scene descriptions with global and local captions. Then the text-only LLM (we use GPT3) will compare the multi-granularity visual descriptions with the input text prompt to give a score according to the evaluation guideline prompt.

**VIEScore** [18] rates aspects of semantic consistency (SC) and perceptual quality (PQ) ultimately providing a rating score on a scale of 0 and 10. We use 0-shot LLaVA-1.5 as the backbone MLLM to evaluate how successfully the image follows the text-to-image prompt.

## 5 Results

Table 2 shows the results for the **Ordering** feature rank$_m$ and **Separation** features sep$_m$ and delta$_m$ for each metric we assessed, on average for all SEGs (Avg), and the three SEG subsets, as well as an approximate FLOP cost per run for each metric (§A.3).

We found that the Synth set consisting of hand-designed (and probably more obvious) errors was the easiest subset for all metrics to correctly order. The average rank$_m$ score for Synth across all metrics was 70%, for Nat 55% and for Real 56%. However, different subsets were hard for different classes of metrics. For the QG/A metrics, Real was hardest, while Nat was harder for the other classes.

As the embedding-correlation metrics came first, TIFA [16], DSG [6], LLMScore [17], and VIEScore [18] all compare themselves against a CLIPScore baseline [15]. Despite the superiority these metrics supposedly held on their respective ad-hoc evaluations, *the computationally cheaper CLIPScore and ALIGNScore are Pareto-optimal in most cases* (Figure 4), sharing the optimality frontier with DSG or TIFA with GPT-4V, methods that are ≈ 6 orders of magnitude more computationally expensive.

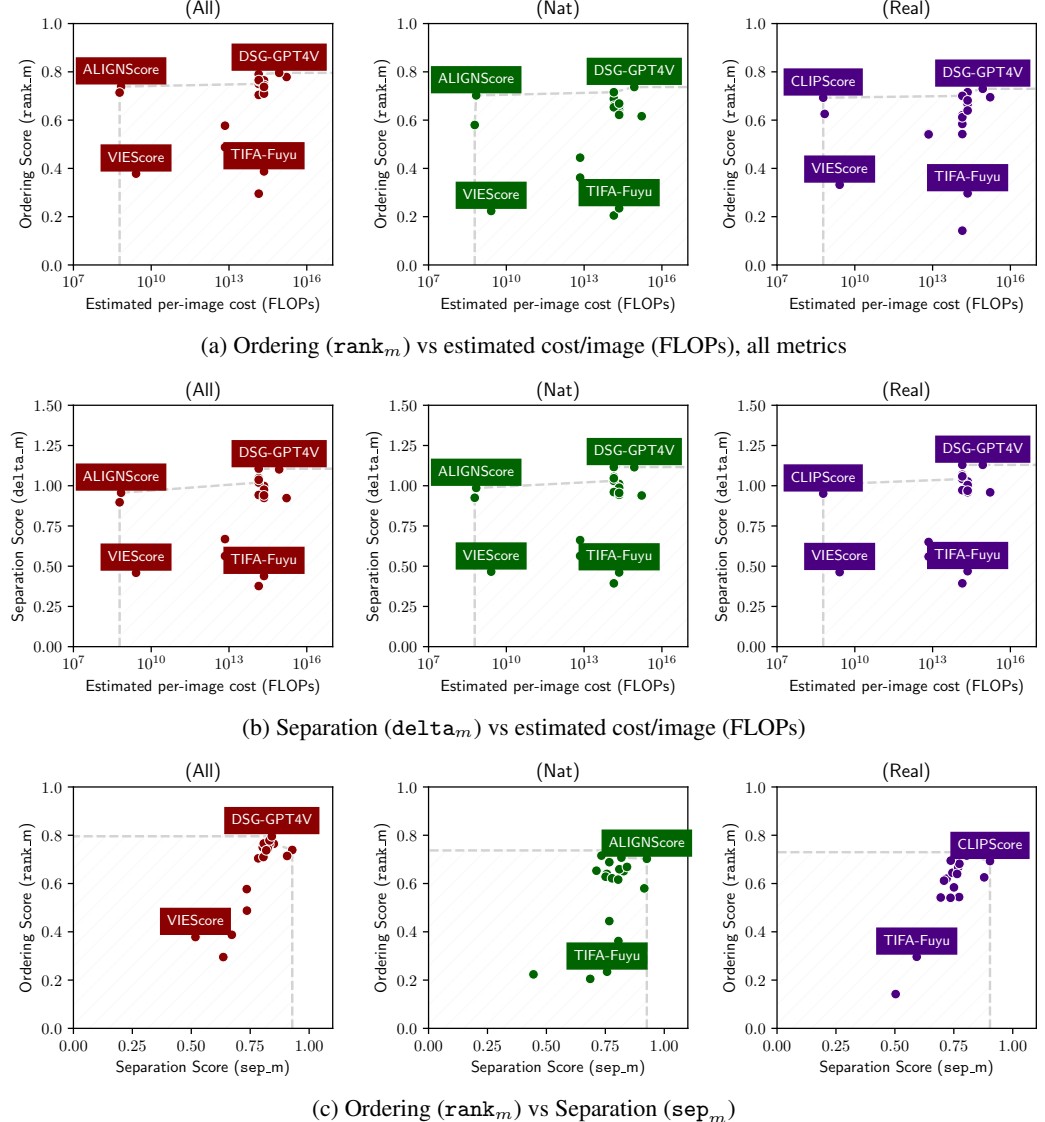

(a) Ordering ($\mathtt{rank}_m$) vs estimated cost/image (FLOPs), all metrics

(b) Separation ($\mathtt{delta}_m$) vs estimated cost/image (FLOPs)

(c) Ordering ($\mathtt{rank}_m$) vs Separation ($\mathtt{sep}_m$)

Figure 4: Plots of ordering and separation scores against estimated per-image metric evaluation costs in FLOPs and each other. For all analyses, the Pareto optimal metrics are DSG and TIFA with GPT-4, and the vastly less expensive embedding-correlation ALIGNScore and CLIPScore.

## 6 Discussion

The headline takeaway from our findings is that, contrary to claims of superiority leveled in their own papers, for all the QG/A and caption-comparison metrics [6, 16–18], *the cheap embedding-correlation metrics such as CLIPScore are sufficient or even preferable* at capturing objective semantic errors relative to fixed prompts. We view this capacity to accurately discriminate similar images relative to a prompt as the core feature a good prompt faithfulness metric must possess.

**T2IScoreScore** is effectively evaluating T2I metrics as relative *score regressors*—functions that are predicting a specific score for an image. However, there are additional desirable elements to a Human aesthetic preferences are—by design—ignored in **TS2** meta-evaluation. Though the QG/A and caption-comparison metrics fail to meaningfully outperform the cheap embedding-correlation metrics, they may have advantages that are not captured by **TS2**, such as in modeling human aesthetic preferences. Paired with a standalone benchmark of human aesthetic preferences over error-free images, a metric's error assessment and aesthetic fidelity could be measured independently.

As the first objective evaluation of faithfulness metrics based on structural semantic errors, **TS2** enables more fine-grained measurement of metric desiderata, leading researchers to build better metrics, and empowering developers to make trade off-informed metric choices.

## 6.1 Pareto frontiers with compute cost

Why do we care about compute cost? When evaluating T2I models at release, compute costs are not very important—an evaluation only has to be run on a small set of images from benchmark prompts.

However, during training or in online monitoring, compute cost for faithfulness metrics becomes quite important. Faithfulness metrics could be used as reward signals while training a T2I model (or prompt generator), or called repeatedly during validation passes. Faithfulness metrics could be deployed in applications to guide an online prompt refinement system, to trigger a second call in user-facing applications, or to analyze prompt corpora to surface challenging examples for further training or analysis. In all of these settings, a performant, low-cost model such as CLIPScore is valuable. **TS2** demonstrates that the performance premium for the expensive metrics is quite small.

## 6.2 Considering error graphs enables objective evaluation

Previous evaluations of coherence metrics have evaluated metrics as human preference score regressors over single images, or over image pairs. The challenge with such an evaluation is that *human preferences are not objective*—especially when provided by a small pool of annotators, correlation to such scores is an unclear signal.

However, by instead evaluating walks over error counts in SEGs, the **TS2** captures a more objective notion of correctness, by ignoring the subjective relationships between pairs of unconnected nodes.

For example, consider the SEG presented in Figure 1, where two different single-error nodes are shown. Given the prompt "a bot in a green shirt poses with some fruit," one of these nodes contains images without fruit, and the other contains boys wearing a blue shirt, rather than green. Which of these types of images are actually worse with respect to the prompt? This is a subjective decision— some annotators may find the missing fruits more important than the incorrectly colored shirt. **TS2** ignores this distinction, as no nodes of equivalent error counts are connected in any SEG. While the difference between those nodes is subjective, the difference between both of these nodes and their shared child node—one where fruits are missing **and** the shirt is incorrectly colored—is objective.

## 6.3 Human baselines and metric ignorance of ranking task

It is also important to note that metrics under test are not aware of the implicit ranking task in **TS2**, as they are evaluated as score regressors. Objective human annotation was only possible *because the annotators were aware that relative ranking was the goal*.

If human performance were judged on the task of simple Likert scoring of image-prompt accuracy without instructions, humans may not significantly outperform the metrics. However, if the human annotators were instructed to count the number of errors, we suspect they would perform quite well, even without the other images for comparison over which the ranking task is performed.

Though we provide no human baseline, we do not think this is a significant weakness—human performance on the inherently synthetic task of image quality scoring is not as important as performance on ranking along objective errors.

## 6.4 Systematic advantages for some metrics

One disadvantage of using Spearman's $\rho$ is that it "expects" ties to be the same in both distributions. For example, if a set of images has error count $(0, 1, 1, 2)$, the ordering $(1, 0.5, 0.5, 0)$ will have a perfect $\rho = 1$, while the ordering $(1, 0.51, 0.49, 0)$ will be penalized, despite it also presenting a correct ordering. This means that **our Ordering score $\text{rank}_m$ systematically punishes the embedding-based metrics relative to the VLM-based ones**, as the embedding-correlation metrics CLIPScore and ALIGNScore can take continuous values, whie TIFA, DSG, LLMScore, and VIEScore have a discrete range. In light of this systematic disadvantage for embedding-correlation metrics, it is even more striking that CLIP/ALIGNScore still are so performant and on the optimality frontier.

## 6.5 Near-perfect performance on $\mathtt{rank}_m$ and $\mathtt{sep}_m$ possible

Although the scores of many models on our metric are high, this meta-evaluation is far from "solved." In principle it should be possible to get much closer to 100 on average for both meta-metrics than we find. We view use of **TS2** as a necessary secondary evaluation for any new proposed T2I faithfulness metric; if it has high correlation to subjective human judgements but does not perform well on **T2IScoreScore**, skepticism might be warranted.

## 6.6 Impact of future VLM advances

Ultimately, all image coherence evaluation metrics stand to improve from further advances in general VLM quality. As a considerably more performant model than LLaVA, mPLUG, etc, it was unsurprising that GPT4-V worked much better as a backbone for TIFA and DSG than the aforementioned. However, there do appear to be diminishing returns, as the order of magnitude going from mPLUG- to GPT4-V-based evaluation yielded a sub-1% improvement in $\mathtt{rank}_m$ performance on the most difficult and construct-valid Real set. Better constraint-generating processes may be required to push VLM-based evaluation metrics further.

## 7 Conclusion

We introduced **T2IScoreScore**, a first-of-its kind objective evaluation for text-to-image faithfulness metrics that utilizes a high image-to-prompt ratio to organize its reference images along semantic error graphs, through which a faithfulness metric can be assessed by our novel graph-based meta-metrics. Our study reveals a surprising finding: more expensive and recent "state of the art" VLM-metrics actually only have modest gains in performance over simpler and cheaper embedding-based metrics at best. Indeed, these cheap metrics such as CLIPScore and ALIGNScore are actually Pareto optimal along with the vastly more expensive and slightly more performant GPT-4V-based QG/A metrics, even when strictly comparing ordering and separation capabilities (leaving compute cost aside).

This underscores the necessity for a more nuanced approach to benchmarking and developing metrics capable of capturing the subtle semantic nuances between prompts and generated images. The establishment of **T2IScoreScore** as a benchmarking tool is a significant step forward, offering a structured way to rigorously test and improve T2I prompt faithfulness metrics, ensuring they can more accurately reflect the semantic coherence between prompts and generated images, thereby facilitating the development of more reliable and effective T2I models.

## Limitations, ethical considerations, and impact

Limitations to our work are discussed throughout. For example, in §6.4 we discuss how our meta-metrics are limited by intrinsic biases of rank-correlation metrics among ties (many of which occur when multiple images occupy one node on a SEG). Additionally, compared to other evaluation sets, our total number of prompts is modest (this is required to achieve a high image-to-prompt ratio, however, which is a core strength of our work). Finally, due to its secretive nature, we are only able to produce *rough estimates of the compute cost of GPT-3 and GPT-4 based metrics*. We estimate them to the best of our ability using third-party information (§A.3).

This research will steer the development of more effective faithfulness metrics, which in turn will guide T2I model development. T2I models are inherently dual-use: they can be used to produce misinformation and other harmful content in addition to useful and entertaining imagery. Any work that contributes to improving their overall performance necessarily drives a small amount of both positive and deleterious impact in this way.

## Acknowledgements

Thank you to the Fatima Al-Fihri Predoctoral Fellowship program for compute support. This work was supported in part by the National Science Foundation Graduate Research Fellowship Grant No. 1650114, CAREER Award Grant No. 2048122, and the Neal Fenzi Resonant Founder Fellowship.

## Contribution Statement

MS checked SEGs, designed the benchmark and meta-metrics, implemented the SEG tree iteration process and evaluation code for the ordering and separation scores, collated the QG/A answers into ID-level scores, and assessed the final scores.

FJ produced and annotated the Synth SEGs, produced and annotated a subset of the Real SEGs, and checked all others. FJ collected answers for Fuyu for the QG/A metrics and cleaned and organized the final dataset release.

MK produced the Nat SEGs and produced, annotated, and checked the other SEGs. MK generated the TIFA and DSG questions for all prompts, implemented and evaluated CLIPScore and ALIGNScore, collected answers for the QG/A metrics from BLIP, InstructBLIP, GPT-4V and refactored code.

YL evaluated LLMScore for the examples and conceived of measuring faithfulness errors in T2I faithfulness metrics. AS collected answers for the QG/A metrics from LLaVA and VIEScore.

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

# A  Supplementary Details

## A.1  Equations for evaluated metrics

**Embedding correlation metrics**  CLIPScore and ALIGNScore are computed using positive cosine similarity between text and image features, from feature extractor model $\mathcal{M}$ as:

$$\texttt{clip-s}(p,i) = \max(\cos(\mathcal{M}_{\mathrm{I}}(i), \mathcal{M}_{\mathrm{T}}(p)), 0) \tag{6}$$

**VQA metrics**  VLM-VQA metrics like TIFA and DSG are assessed by a multiple choice question assessment model ($\mathcal{M}_B$ and a vision-language model $\mathcal{M}_{VL}$ over questions $Q$ generated by an LLM based on the prompt $p$.

$$\texttt{tifa-s}(p,i) = \frac{1}{|Q|} \sum_{(q,a)\in Q} \mathbb{1}(\mathcal{M}_B(\mathcal{M}_{VL}(i,q)) = a) \tag{7}$$

## A.2  VLM details

**mPLUG** is a class of vision-language models that use skip connections between visual encoder embedding layers between cross-modal attention blocks in the transformer stack [46]. We use the mPLUG-OWL [47] 7b checkpoint which uses LLaMA 7b [48] as the pretrained text encoder.

**LLaVA** is a fine-tune of Vicuna [49] (decoder-only transformer model) that uses a learned MLP "vision-language connector" layer to map a single input image's CLIP encodings into a shared embedding space [50, 51]. We use LLaVA 1.5 13b. Because LLaVA was instruction fine-tuned for chat applications, we experiment with a variant system prompt that requests *concise* answers from the system. We mark this alternate option LLaVa 1.5 (alt) in plots and figures.

**BLIP** is a jointly-trained self-attention ViT trained with cross attention to multiple transformer encoder and decoder pipelines with different tasks [52]. We use the BLIP encoder/causal LM decoder combination as a transformer encoder-decoder model to produce VQA answers from the `blip-vqa-base` checkpoint.

**InstructBLIP** extends BLIP by including an instruction fine-tuned "Q-Former" that selects salient instruction-related visual features from a frozen ViT for input to a frozen LLM that answers the query conditioned on the selected features [53]. We use `instructblip-flan-t5-xl`.

**Fuyu** is a decoder-only VLM that splits an input image into a sequence of patches that are separately projected directly into the transformer embedding space, which jointly learns ViT and LM behaviors [54]. We use `Fuyu-8b`.

**GPT4-V** is the largest state-of-the-art VLM provided by OpenAI [55]. It is expensive to run!

## A.3  Compute cost estimates

**Estimating FLOPs per inference pass for each model.**  We use OpenAI's estimate [56] of $\approx 2N$ OPs per forward pass for a large transformer model, where $N$ is the total number of parameters.

**Obtaining parameter count estimates for closed models.**  TIFA, DSG, and LLMScore use some combination of GPT3 and GPT4, whose exact parameter counts and FLOP/inference costs haven't been publicly disclosed. We use estimates from SemiAnalysis to get an approximate FLOP cost. While these numbers are likely imperfect, their orders of magnitude are as accurate as we can get.

**Obtaining metric FLOP cost per single image eval.**  Given FLOP/forward pass estimates for each model, our estimates of the FLOP cost to evaluate an image is a function of the number of model calls and estimated tokens per call. The embedding-correlation metrics require 2 calls to an embedding model, matmul and sum operations to get the cosine similarity. TIFA requires on average 8 questions, with GPT-3 question-generating calls of average 40 tokens, and VQA model calls of average length 20 tokens. For DSG these numbers are 5, 40, 15. The costs of calling the freeform-to-multiple choice model are negligible. We estimate that LLMScore and VIEScore both require approximately

50 tokens of LLM or VLM compute to score an image. LLMScore's use of mPLUG to caption is negligible alongside the cost of running GPT-3.

**Total compute cost of our study.** In total, we estimate our study took $9.89 \times 10^{18}$ FLOPs, mostly through OpenAI's service, but also on lab-owned NVIDIA Titan-X and A-100 GPUs.

## A.4 Semantic Error Graph Structure

For more information on the structure of the semantic error graphs (SEGs), we provide examples here. SEG 85 is one example with a more interesting topology than the example in Figure 1. Figure 5 has a structure including two-error edges, single-parent nodes, single-child nodes, multi-parent nodes, and multi-child nodes in the same graph, corresponding to prompt "*guy with umbrella hat sitting at a table with another person with a hat under a red umbrella.*"

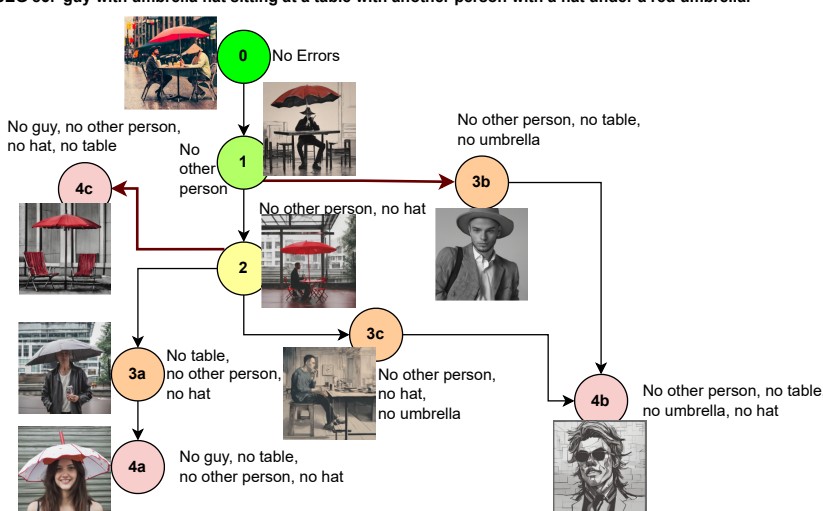

Figure 5: Example of a SEG (85) with a more complex structure. Some nodes have multiple child nodes, and some edges correspond to more than one error (**dark red**).

## A.5 Scoring within SEGs

Figure 7 exemplifies why we choose to only score rank order *along walks* of the graph, rather than *between all pairs of nodes*. A priori there's no reason the beach-less images should be worse than the umbrellas, yet metrics consistently rate the beach error more severe.

## B Supplementary Results

### B.1 Comparing DSG question evaluation using DSG and TIFA score accumulation methods

The first few steps of TIFA [16] and Davidsonian Scene Graph (DSG) [6] scoring methods are nearly identical: an LLM generates a set of requirements as questions, and a VQA system answers them. However, the two methods differ chiefly in how the answers are combined into a single image-level score. TIFA simply scores images by the correct answer rate, while DSG uses the graph structure of the requirements to build in some robustness: if an *upstream* requirement is not met (e.g., *is there a boy?* : **no**), then *downstream* requirements are all also assessed as not being met, regardless of answer. In the example provided, if the question "*is the boy's shirt green?*" were answered **yes**, the DSG accumulation technique would still score this requirement as being not met, due to the upstream requirement, while the TIFA accumulation method would score it as being met.

**SEG 109: a Mesoamerican pyramid surrounded by jungle. Detailed charcoal sketch.**

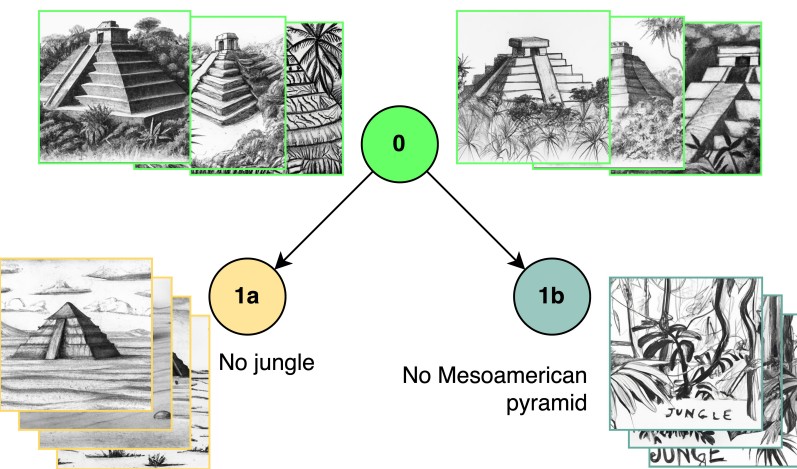

Figure 6: Example of a SEG (109) with a simpler structure. We show multiple images for each of the three nodes in this SEG.

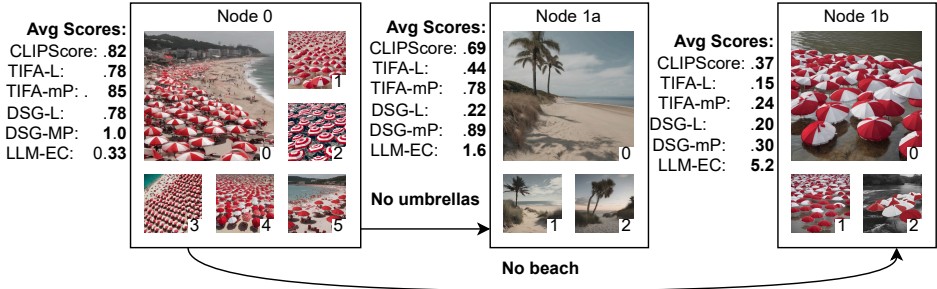

Figure 7: Examples from SEG 71 (*The beach is crowded with red and white umbrellas*). Even though both nodes 1a and 1b have the same error count (1) they systematically differ across all metrics: all metrics punish the images where the umbrellas are just in water (no beach, 1b) more than they penalize an empty beach with no umbrellas (1a).

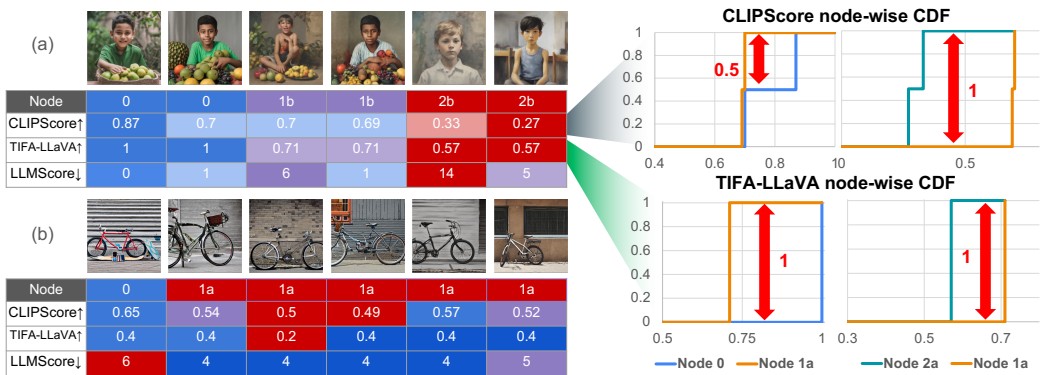

Figure 8: Examples of scores assigned by three metrics to examples from an easy (a) and hard (b) semantic error graph (left). Computation of the separation score $\mathrm{sep}_m(\mathrm{S})$ for two metrics is depicted at the right. Color coding of each cell corresponds to the metric's score for the image being better (blue) or worse (red); more correlated measures (presenting a higher rank order score $\mathrm{rank}_m(S)$) will show the same progression from red to blue (a), while harder-to-rank examples will not (b).

| | | mPLUG | LLaVA 1.5 | LLaVA 1.5 (alt) | InstructBLIP | BLIP1 | Fuyu | mPLUG | LLaVA 1.5 | LLaVA 1.5 (alt) | InstructBLIP | BLIP1 | Fuyu |
|---|---|---|---|---|---|---|---|---|---|---|---|---|---|
| | | | | DSG w/ TIFA accumulation | | | | | | DSG w/ DSG accumulation | | | |
| **Ord.** | Avg | 70.4 | 76.2 | 75 | **79.0** | 76.6 | 29.5 | 68.8 | **80.0** | 75.6 | **80.2** | 76.9 | 35.8 |
| | Synth | 74.6 | 80.1 | 81.6 | **85.1** | 81.6 | 35.4 | 73.5 | 83.8 | 82.1 | **86.1** | 81.7 | 45.5 |
| | Nat | 65.3 | 65.9 | 68.8 | *70.7* | **71.6** | 20.5 | 61.9 | **74.9** | 68.9 | 70.2 | 71 | 21.5 |
| | Real | 58.4 | **70.0** | 54.2 | 62 | 61.2 | 14.2 | 56.4 | **69.6** | 55.9 | 65.8 | 62.8 | 10 |
| **Sep.** | Avg | 78.4 | 83.1 | 80.3 | 84.2 | 80.8 | 63.6 | 75.5 | 82.5 | 80.5 | 84.3 | 80.6 | 66 |
| | Synth | 80.9 | 85.7 | 83.9 | 87.8 | 84.9 | 65.8 | 77.1 | 85.5 | 83.8 | 88.8 | 84.1 | 68.7 |
| | Nat | 71.2 | 80.9 | 76.7 | 81.8 | 73.3 | 68.6 | 70.6 | 75.1 | 77.2 | 81.5 | 75.1 | 71 |
| | Real | 75.1 | 74.5 | 69.4 | 71.9 | 70.8 | 50.3 | 73.1 | 76.8 | 70.6 | 68.9 | 71.4 | 50.8 |

Table 3: Comparing how using DSG vs TIFA-style accumulation for scoring each image by DSG questions impacts performance along both our metrics. The right half of this table is identical to the DSG section in Table 2, and bold, italic, and highlighting follows the same rules, except cells in the TIFA half are marked as if they were replacing the right half cells in the DSG section in Table 2.

As a supplementary experiment, we compare how accumulating the DSG questions using the DSG technique compares to accumulating them with the TIFA technique in Table 3. Interstingly, the impact of this change differs between strong and weak VLMs, between ordering and and separation scores, and between the easier and harder subsets. For example, switching from the DSG to TIFAstyle acculumation consistently *improves ordering performance for mPLUG*, while it *worsens performance for LLaVA, InstructBLIP, and BLIP1*. For Fuyu, the weakest model, DSGstyle accumulation *significantly improves performance* over TIFA. This strengthens the claim from [6] that using the scene graph to check requirements adds robustness; it makes a lot of sense that this robustness benefits the lowest-performing VQA systems the most.

For separation scores, TIFA accumulation improves performance of more models. In particular, TIFA accumulation pushes InstructBLIP into the top 3 for separation on the Synth subset, while no DSG metric using DSG accumulation breaks into the top 3 (red highlighted cell).

## B.2 Modelwise Spearman Ordering Score Histograms

Here we provide full histograms for our Spearman Ordering and Kolmogorov–Smirnov Separation scores, across every SEG, for all metrics we assessed.

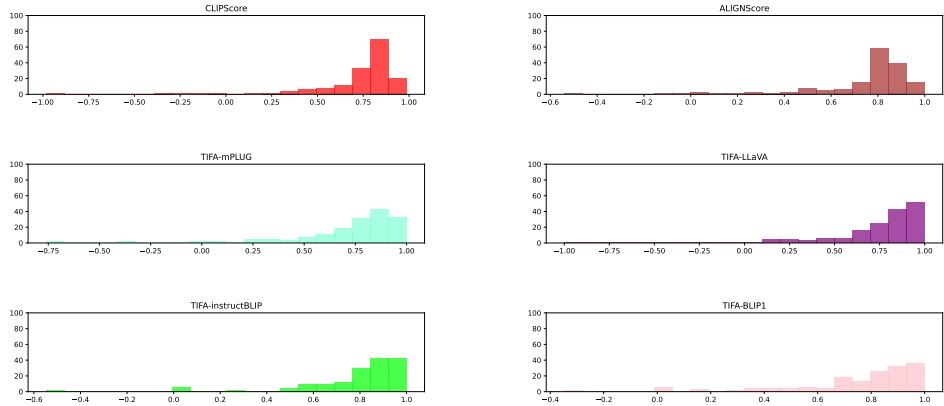

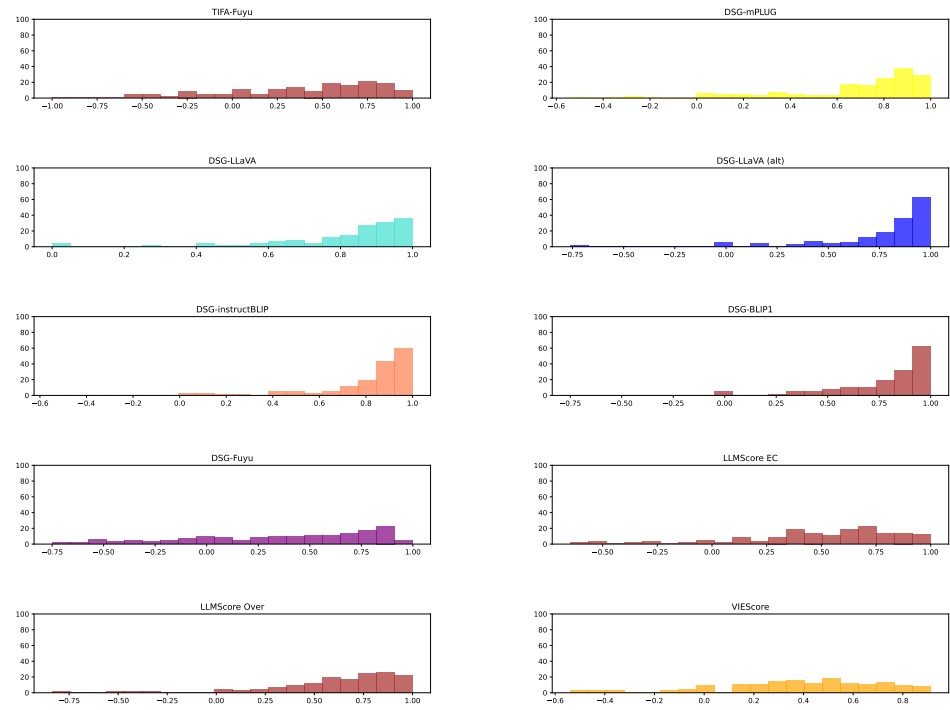

## B.3 Modelwise K–S Separation Score Histograms

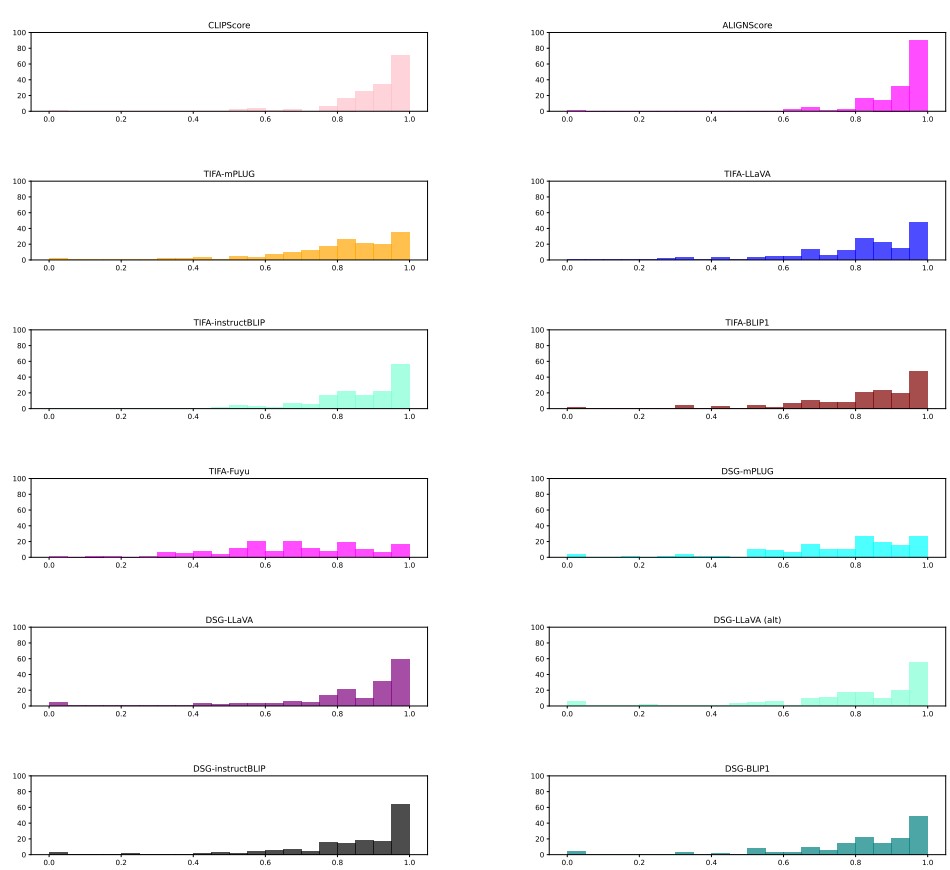

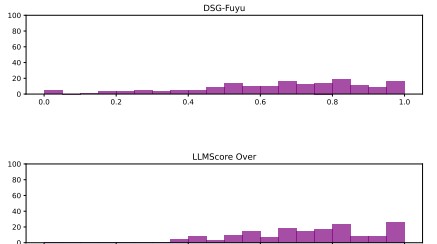

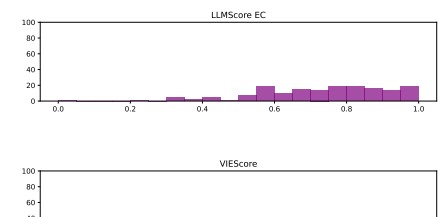

Here we provide line plots for a set of metrics and SEGs. Note that for `normalized_rank`, higher is worse (more errors). High-correlation is assessed when the metric lines (higher better) go down as the metric lines go up.

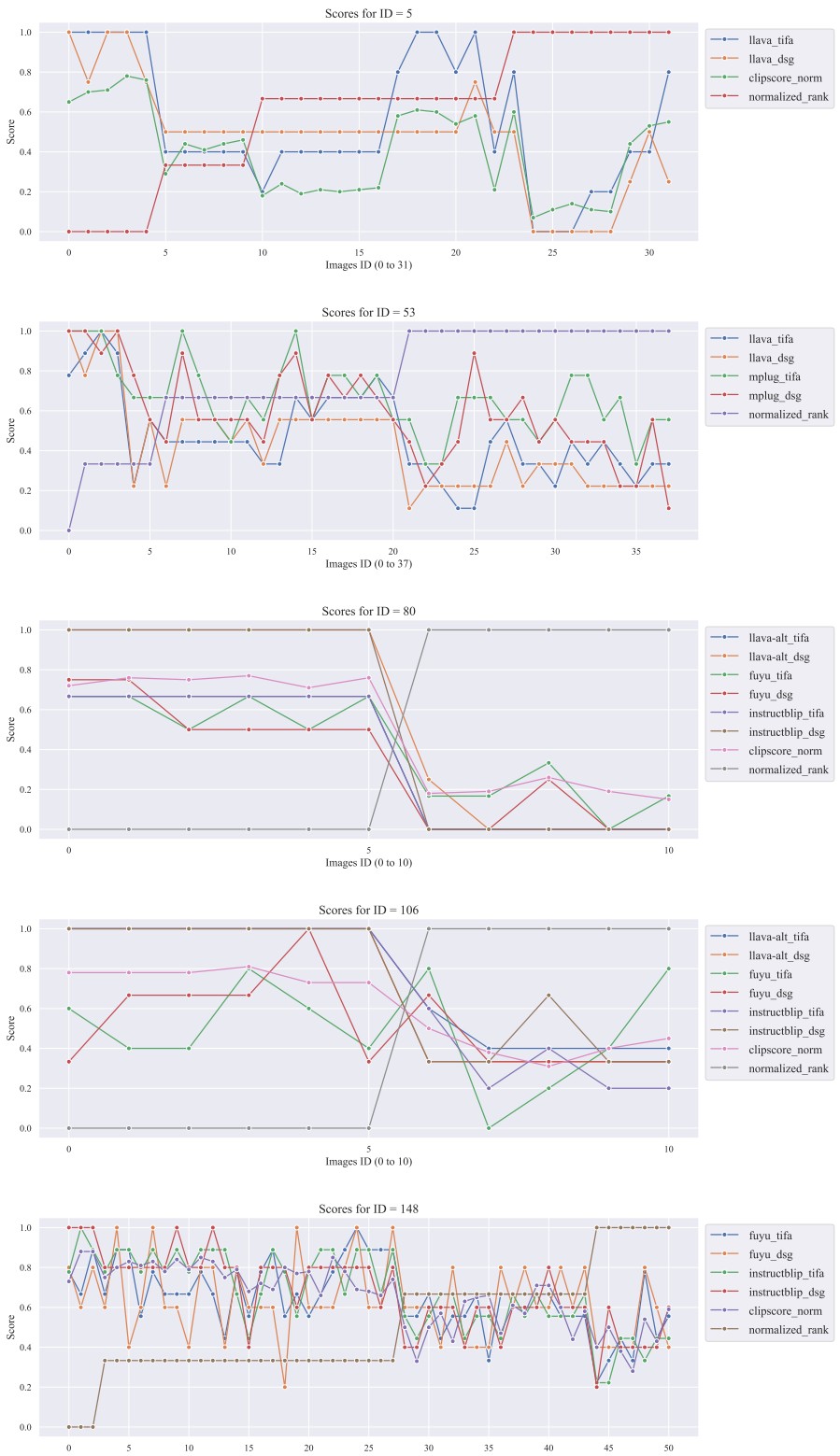

# C  Supplementary Analysis

Another interesting weakness of the QG/A metrics is that many unlucky situations where the VLM backend presents a mix of true and false positives that cause incorrect rankings or poor separation (DSG fails to order samples while CLIPScore succeeds in fig. 9) to occur. However, these VLM failures cases are interpretable and can be targeted; `T2IScoreScore` will hopefully drive future work in making LMs more robust to these sorts of errors for VQA to mitigate this issue. In addition to these interpretability advantages, the more sophisticated VLM-based metrics still do present better subjective human preference correlation than CLIPScore [6, 16–18]. By focusing exclusively on objective similar-image ordering and separation, `TS2` is effectively orthogonal to these preference evals.

Given the documented biases LLMs have in directly outputting numbers [57, 58], it isn't a surprise that the technique which directly prompts VLMs to output a numerical preference value (VIEScore) is at present the least robust.

In general it seems that the most successful methods that leverage VLMs (TIFA and DSG) still ultimately produce scores using a deterministic algorithm. They use VLMs in a perceptual manner to separately check each requirement, but the final score is the accuracy estimate from each separate VQA question. This comports with the theories of LLM function that treat it as a "system 1" [59] ; effectively TIFA and DSG are examples of *VLM-modulo* frameworks outperforming pure LLMs on the task of prompt coherence scoring [60].

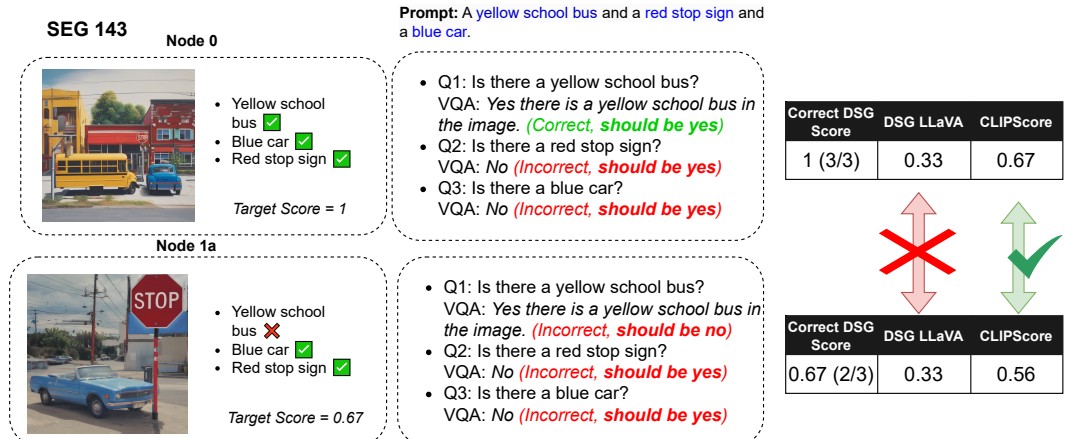

Figure 9: Example of two images on nodes 0 and 1 from a hard SEG that are correctly separated (and ranked) by CLIPScore but are not separated by DSG-LLaVA. VLM hallucinations are a key hinderance to QG/A performance on `TS2`.

Are the same SEGs hard for the same models? fig. 10 and fig. 11 present correlation plots between SEG-wise $\text{rank}_m$ and $\text{sep}_m$ scores respectively between each pair of metrics. For both we show (a) the correlations over all SEGs, and (b) the correlations between only SEGs in the Real subset. These plots show that broadly, similar methods have similar "blind spot" SEGs, while different ones can vary wildly in terms of which examples they succeed and fail at ordering and separating. Note that all TIFA or DSG QG/A metrics have appreciable correlation to each other, provided they use a strong enough VLM. The metrics employing weak VLMs such as Fuyu do not perform well. Similarly, the two LLMScore metrics are highly correlated to each other; the pure VLM numerical rating methods are not producing random noise. These correlations are stronger in the full set of SEGs (including natural images and the easy, pre-designed Synth SEGs) than they are in the hardest Real set of SEGs.

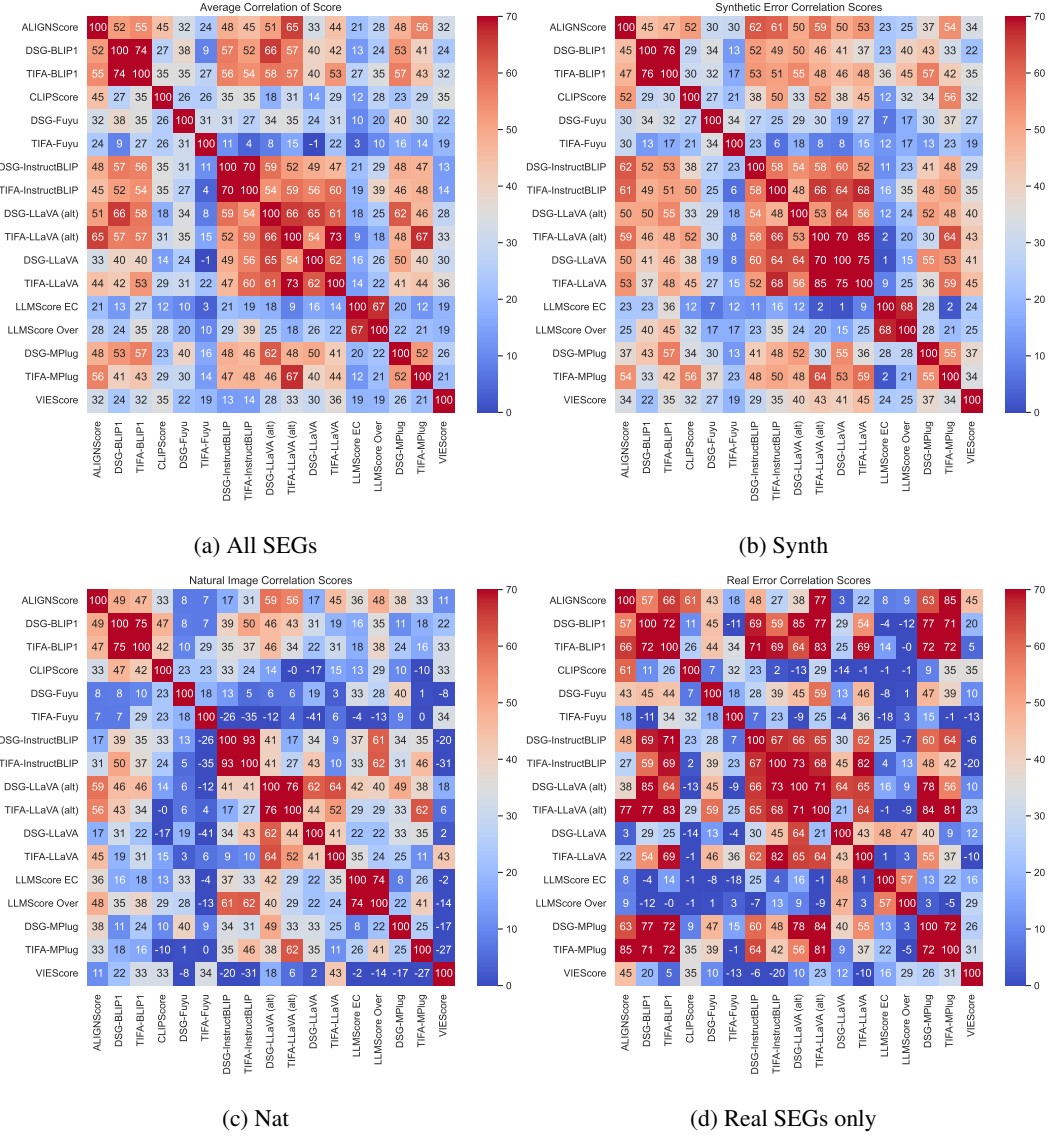

(a) All SEGs

(b) Synth

(c) Nat

(d) Real SEGs only

Figure 10: Correlation between the Spearman correlation score for each prompt tree for each metric, for all SEGs (a), for the synthetic error SEGs (b), for the natural image/synthetic error SEGs (c) and for the real error subset (d).

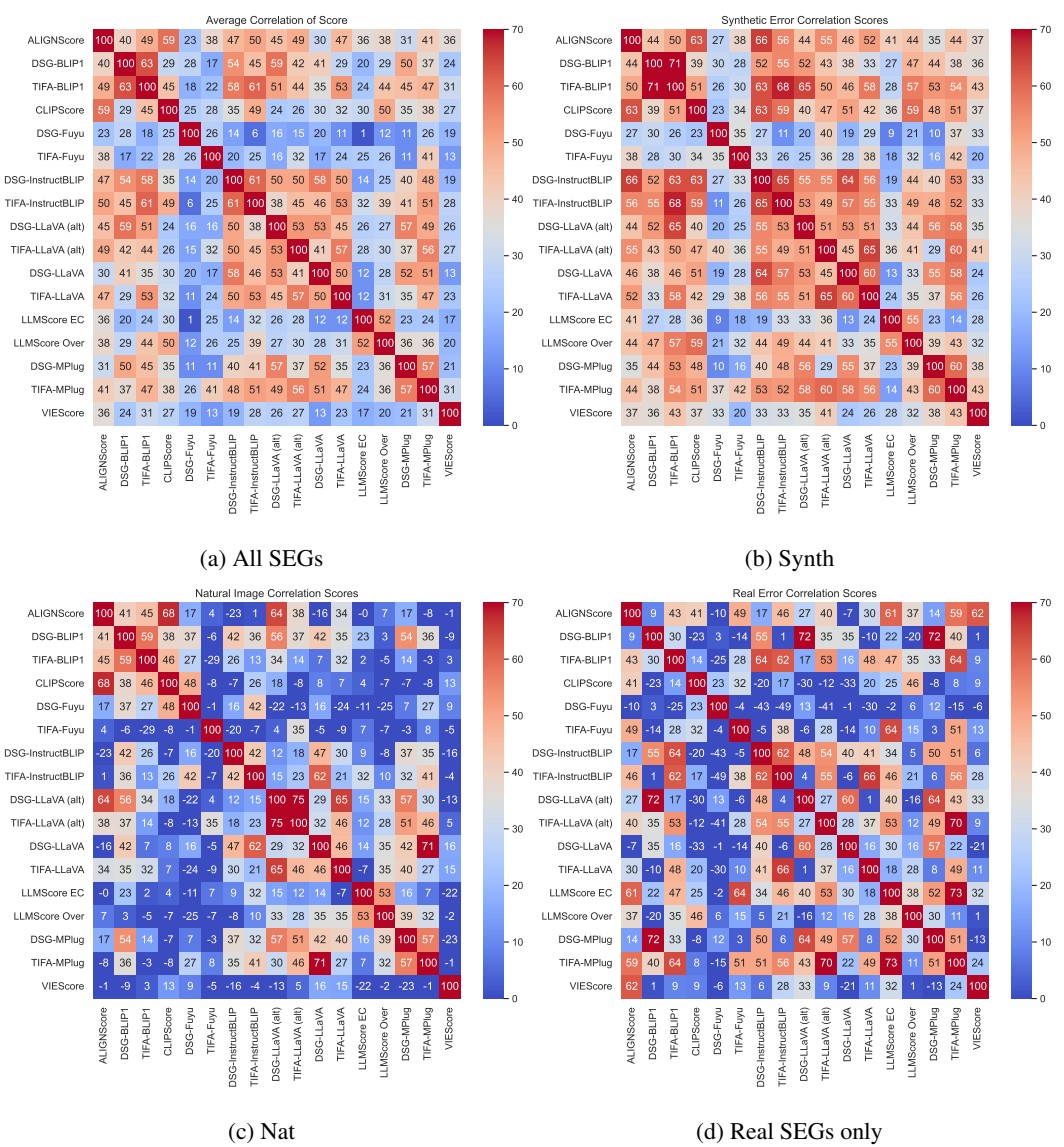

(a) All SEGs

(b) Synth

(c) Nat

(d) Real SEGs only

Figure 11: Correlation between the K–S Separation score for each prompt tree for each metric, for all SEGs (a), for the synthetic error SEGs (b), for the natural image/synthetic error SEGs (c) and for the real error subset (d).

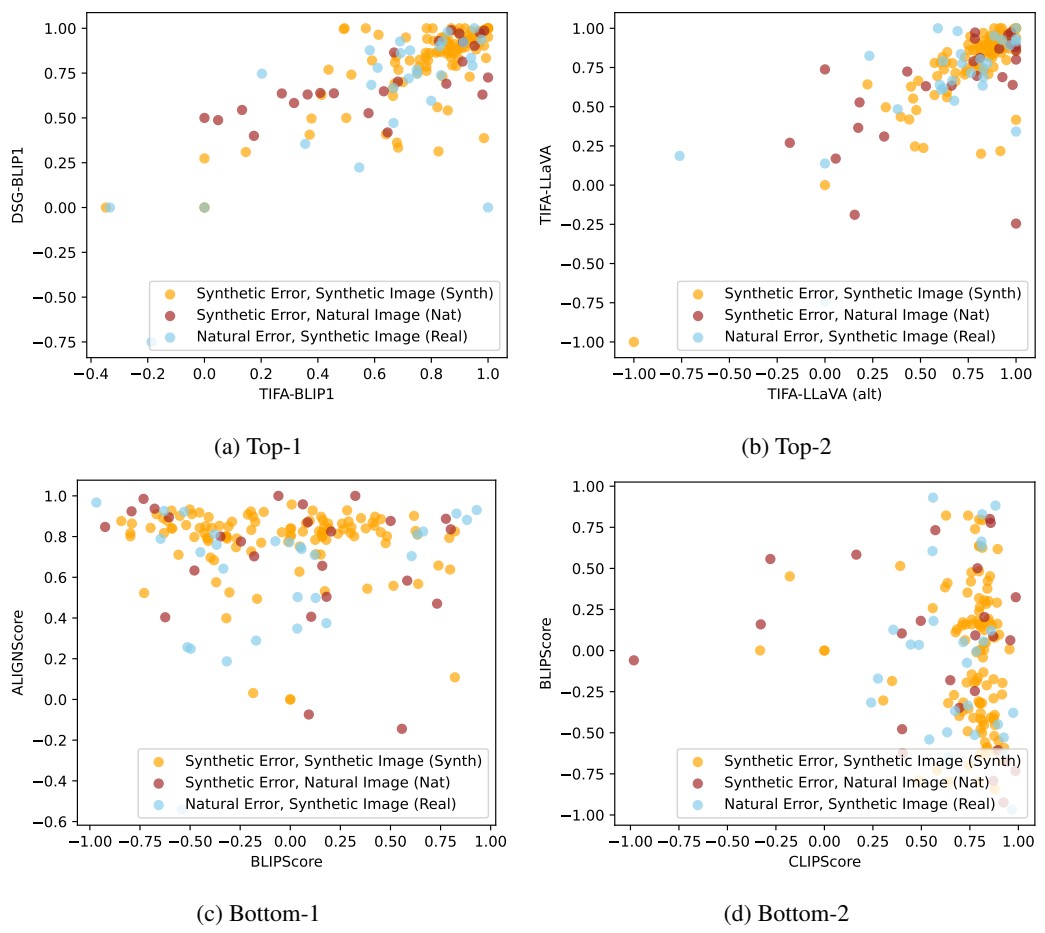

(a) Top-1

(b) Top-2

(c) Bottom-1

(d) Bottom-2

Figure 12: Scatter plots comparing the two most correlated metrics (a, b) by Spearman correlation Ordering score across the Synth, Nat, and Real populations, and the two least-correlated (c, d). Note that the two highest-correlated metrics are both QG/A metrics using the same underlying VLM (DSG and TIFA using BLIP1, (a); TIFA using LLaVA with two different system prompts, (b)).

Figure 12 and Figure 13 show scatter plots for the Ordering (Spearman) and Separation (KS statistic) scores for every SEG between the most highly-correlated (a, b) and low-correlation (c, d) pairs of metrics under evaluation, respectively.

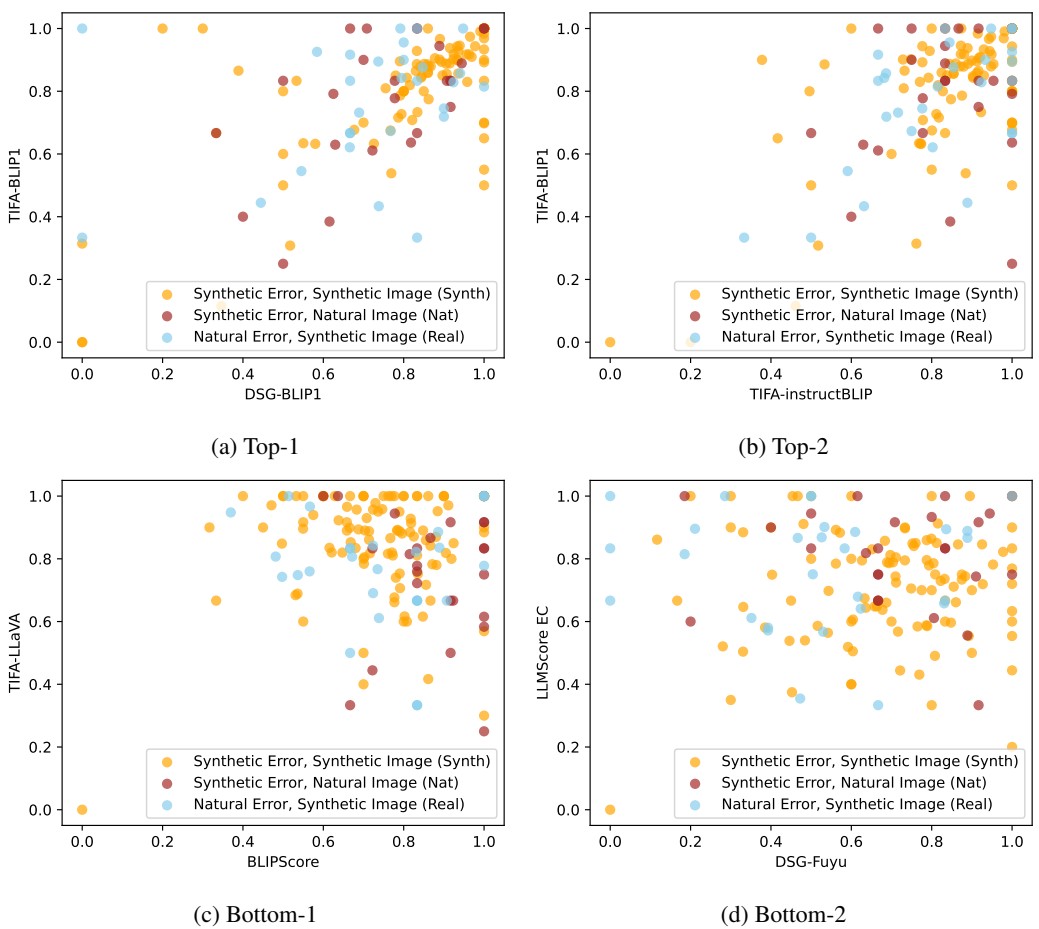

(a) Top-1

(b) Top-2

(c) Bottom-1

(d) Bottom-2

Figure 13: Scatter plots comparing the two most correlated metrics (a, b) by Kolmogorov–Smirnov Separation score across the Synth, Nat, and Real populations, and the two least-correlated (c, d). Note that the two highest-correlated metrics are both QG/A metrics using the same or related underlying VLMs (DSG and TIFA using BLIP1, (a); TIFA using BLIP1 and InstructBLIP, (b)).

Both of these sets of figures confirm that similar underlying VLMs by-and-large "think" similarly in terms of scoring models, even over different sets of questions (TIFA and DSG). This suggests that development of overall better VLMs will generalize to many different types of VLM evaluations.

