# OpenReview forum: "Who Evaluates the Evaluations? Objectively Scoring Text-to-Image Prompt Coherence Metrics with T2IScoreScore (TS2)"
_NeurIPS.cc/2024/Conference — NeurIPS 2024 spotlight_

### Official Review · Reviewer_Xy9h · 2024-07-07

**Soundness:** 3
**Presentation:** 3
**Contribution:** 3
**Rating:** 6
**Confidence:** 4

**Summary:**

The authors propose T2IScoreScore (TS2), a benchmark and set of meta-metrics for evaluating text-to-image (T2I) faithfulness metrics. Compared to existing relevant benchmarks, TS2 has higher image-to-prompt ratios, which allows users to organize semantic error graphs (SEGs), where each edge corresponds to a specific error with respect to the prompt that a child image set possesses but its parent images do not. Based on SEG, the authors evaluate T2I metrics, including embedding-based (CLIPScore/ALIGNScore), QG/A-based (TIFA/DSG), and caption-based (LLMScore/VIEScore) metrics, by how the metrics properly order and separate images. Different metrics show different advantages, and the authors highlight that simple embedding-based metrics outperform other more computational metrics in separation criteria.

**Strengths:**

1. Introduction of meta-metric benchmarks of recent T2I metrics, including collection of large image-text pairs and semantic error graphs.

2. Comprehensive experiments, including using different VLM backbones for TIFA/DSG.

**Weaknesses:**

**1. Simply treating QG/A metrics as score regressors.**

One of the major motivations behind using QG/A metrics, even though they are computationally expensive, is that they divide multiple aspects of prompts and provide comprehensive skill-specific performances of the T2I model. While the QG/A metrics can be used as a score regressor, comparing them with embedding-based metrics ignores their biggest advantages. This background and limitation needs to be clarified in introduction section, otherwise this could misleading readers who are new to T2I metrics.

**Questions:**

See weaknesses

**Limitations:**

There is no significant negative societal impact.

---

> ### Author Rebuttal · Authors · 2024-08-07
>
> Thank you for your thoughtful review. We are heartened to hear that you view our meta-metrics as a novel strength, and that you appreciate our comprehensive experiments that really attempted to make the strongest possible case for the QG/A metrics by analyzing multiple backends. We hope we can satisfactorily answer your question.
>
> - **On the weakness of evaluating T2I faithfulness metrics as score regressors**
>
> You are right to point out that one of the motivations of the QG/A metrics, which we do not inspect in detail here, is that they can provide fine-grained, (ideally) explainable assessments of T2I outputs over defined axes, as opposed to the other metrics providing simple numerical outputs. It is important to note this as a strength.
>
> However, we are not the only ones who give the QG/A metrics a score regressor treatment—**in their own works introducing the QG/A metrics, the authors justify their contributions with the same score regressor methodology, against the very same correlation-based metrics we analyze**. In their works, using their ad-hoc evaluation sets and simple metrics, they claim significant gains in this score regression performance over the correlation metrics.
>
> Simply by building a more carefully considered test set and metrics that are tailored to evaluate **objective, relative errors alone** without contamination by other forms of human preference such as aesthetics, we find surprising contrary results—that the regressor improvements claimed in the works introducing TIFA and DSG are illusory when considering structural correctness of the images alone.
>
> That being said, you are right to point out that contextualizing the exact implications of our findings is crucial for inexperienced readers, and we will do so. In our camera ready, we will state the following points regarding the purpose of our evaluation:
> 1. TS2 is intended to provide a meta-assessment of metric quality **that isolates structural correctness in relative comparison of related images**
> 2. TS2's evaluation **is orthogonal to human aesthetic preferences,** and that a comprehensive meta-evaluation should also take those preferences into account
> 3. That our findings, *while important, surprising, and soundly demonstrated,* do not tell the full story of T2I metric quality; **some metric strengths such as explainability and contextualization cannot be captured numerically**
>
> We appreciate you bringing this point to our attention and look forward to using it to strengthen our camera-ready.

---

> > ### Comment · Reviewer_Xy9h · 2024-08-12
> >
> > I appreciate authors' response and decided to keep my current score.
> > Please make sure to incorporate the points you mentioned in the next version if accepted.

---

### Official Review · Reviewer_h3XZ · 2024-07-13

**Soundness:** 3
**Presentation:** 3
**Contribution:** 3
**Rating:** 7
**Confidence:** 4

**Summary:**

This paper presents a rigorous evaluation for text-to-image alignment metrics. This is primarily done by introducing a dataset with several images for each prompt, allowing the construction of semantic graphs that can be used to measure the accuracy of the alignment metrics. From the analysis on the benchmark, a major conclusion is that CLIPScore provides an excellent tradeoff (or is at least on the pareto-optimal frontier) between speed and alignment. VQA-based metrics (e.g TIFA, DSG) while improving over CLIPScore in many cases, come with much higher costs (in some cases orders of magnitudes higher), highlighting important considerations for text-image alignment methods.

**Strengths:**

The dataset collected in the paper is quite valuable, and would be useful for evaluating text-image alignment metrics in the future. The methodology in the paper also seems quite sound to me. I also think the analysis in the paper is very sound, highlighting the cost of running the evaluation metric is an important aspect which is often missed in these methods. The paper is also written very clearly, and is easy to read.

**Weaknesses:**

In terms of models/methods evaluated, I see 2 notable omissions: human-preference models such as ImageReward would also be a good addition since they might also capture some notion of text-image alignment while also cheap to use. Another good addition would be VQAScore (much more recent), but seems to show extremely strong results on several image-text matching benchmarks, while not being nearly as expensive as the Question-Generation methods (e.g. TIFA, DSG).

Minor: While I totally agree with the issue of methods evaluating on their own proposed test set, the column "ad-hoc" in Tab. 1 makes little sense without an explanation (line 85 seems too limited). Either a more complete explanation should be given, or it would be better to replace this column with something more objective/concrete.


[a] Lin et al. "Evaluating Text-to-Visual Generation with Image-to-Text Generation", 2024

**Questions:**

Overall, I really like the paper and would like to see this accepted. I have a couple of questions for my curiosity.
While 2.8k prompts is still a lot more than TIFA, DSG, is this size still not too small/limited diversity to be able to comprehensively make conclusions about alignment methods?

Is there any idea of what 'human-performance' would look like on these benchmarks, and if existing models are very far off from that?

**Limitations:**

No major concern.

---

> ### Author Rebuttal · Authors · 2024-08-07
>
> Thank you for your detailed review! We appreciate your praise of our work’s *rigorous evaluation*, *valuable dataset contribution*, *sound methodology and analysis*, and *clear writing*. **We are overjoyed to hear that you really like [our paper] and hope to see it accepted.** These are all very heartening! We hope you will find our responses to your questions satisfactory.
>
> ### Weaknesses
>
> - **Omitted metrics**
>
> ImageReward did compare itself against CLIPScore and BLIPScore in its introductory paper, so it is reasonable to consider evaluating it in our study. We decided not to and to instead focus on the correlation metric vs. VQA metric angle, and because it is simultaneously attempting to measure the orthogonal characteristics of image aesthetic quality and image-prompt faithfulness as mediated and combined by human preference annotators. You raise a good point that it still can be evaluated using TS2, and that the findings from this experiment may have interesting implications for the trade-off between capturing aesthetic preferences and prioritizing structural correctness. We will discuss this possibility (although we do not have results for it) in the future work section, and investigate including ImageReward in our final leaderboard that will be linked in the camera ready.
>
> VQAScore is very interesting, and we did see the paper late into the writing process, and before its source code was fully ready for our use. While we are unable to evaluate it in the paper, we do look forward to adding it to our leaderboard. Though we do not have results for it, we will include a reference to it in the camera ready.
>
> - **Explanation of Ad-hoc evaluations**
>
> Yes, we should clarify, the reason we define “ad-hoc” evaluation benchmarks as test sets that were released alongside proposed methods, and call them out in the discussion of related work, is because of the potential concern that these benchmarks—showing the superiority of the metrics they are introduced alongside—may contain some bias (probably unintentional) in favor of the proposed metric they are motivation, and that these benchmarks are not the primary contributions of their introductory works, and thus have less documented design considerations and production methods. By centering the design considerations of our evaluation in this paper, and *not* introducing a new metric, we are able to focus on the rigor of our evaluation, identify weak points in prior evaluations, and as neutral arbiters demonstrate our surprising finding.
>
> ### Questions
>
> - **Regarding size of TS2**
>
> We do believe the size of the dataset is sufficient to defend our findings. In particular, the high **total number of comparisons** that TS2 enables is its primary strength. By organizing the images along semantic error graphs containing multiple walks, the images are able to be reused much more efficiently than in other datasets, to analyze how well models can order them along different sets of accumulated semantic errors. For example, in figure one the image “1-2.jpg” is simultaneously used to check if a metric attends over a boy missing from the picture, as well as if fruit is missing, by comparing it to both “2-0.jpg” and “2-1.jpg”. This effect, applied over all the SEGs, greatly amplifies the utility of each image.
>
> - **Notions of human performance**
>
> Though our dataset was produced through human annotation with a high annotator agreement, it is important to note that **the human annotators were aware of the implicit ranking task, while the metrics under test are not.** We do not think this is a significant weakness of the work, as human performance on the inherently synthetic task of image quality scoring is not as important as performance on ranking along objective errors.
>
> Thus, if human performance were judged on the task of simple Likert scoring of image-prompt accuracy without instructions,  humans may not significantly outperform the metrics. However, if the human annotators were instructed to count the number of errors, we suspect they would perform quite well, even without the other images for comparison over which the ranking task is performed.
>
> Thank you for this stimulating idea, we look forward to including this discussion in our camera ready.

---

> > ### Comment · Reviewer_h3XZ · 2024-08-09
> > **Some Comments/Suggestions**
> >
> > I thank the authors for their reply, I have no major concerns left about the paper, and I see that all the other concerns of the reviewers are satisfactorily addressed. That said, I have a few comments that the authors may wish to think about:
> >
> > 1) Human-Preference Models: I agree that human-preference models are not immediately clear about what exactly they evaluate. They also start from CLIP/BLIP models, and then finetune it on data which capture some mixture of visual quality (i.e artifacts), aesthetics, and prompt following all at once. For instance, if a vital object from the prompt is missed in the image, the user would naturally rate it low. Similarly, if the image has a lot of artfiacts but follows the prompt well, it is unlikely to do well on comparisons. Depending on the guidelines, annotation protocol, you can get models that result in very different behaviors. For instance, in the VQAScore paper (disclaimer: I have no connection to it), ImageReward outperforms both CLIP, BLIP on most benchmarks (Tab. 4) and is even doing reasonably well on Winoground, Eqben (Tab.3 ). Therefore, I would not dismiss them as solving an entirely different task, and adding TS2 as an additional eval benchmark for these models would be a good idea.
> >
> > 2) Human-Evaluations: I think the authors make an excellent point that humans doing Likert scoring of image-prompt accuracy might actually not outperform existing metrics. This is a useful pointer (since some prior works recommend Likert scoring of image-prompts as a good strategy to evaluate models[a]) in performing more rigorous user studies for evaluating text-to-image models.
> >
> > 3) QG/A metrics doing more than a single score regression: To reviewer Xy9h, the authors point out that QG/A metrics are proposed claiming superior correlation with human judgement on various benchmarks. While I agree with this, the authors should look at Fig. 1 of TIFA which clearly makes the claims of "fine-grained", "accurate", "interpretable". Of course, the fine-grained/interpretable aspects are the hardest to evaluate and justify, therefore papers will inevitably resort to maximizing performance/correlation on benchmarks to justify the method. That does not mean the other aspects of the method are invalid/absent, they are just insufficiently evaluated (beyond a few qualitative examples). Therefore, I would suggest the authors that they acknowledge the strength of QG/A methods, while providing a fair assessment of their shortcomings (which is already there in the paper).
> >
> > I hope the authors can think about these aspects and make the additions/modifications that they deem fit for the camera ready/benchmark leaderboard.
> >
> > [a]: Otani et al. "Toward Verifiable and Reproducible Human Evaluation for Text-to-Image Generation", CVPR 2023

---

> ### Author Response · Authors · 2024-08-09
>
> Thank you for clarifying. These are great points:
>
> 1. TS2 + and orthogonal quality-only eval might actually tease out the degree to which human annotators attend over each consideration by directly comparing their preference correlations to each metric under different annotation schemes. A really cool idea might be to treat those two considerations as principal components a metric could be interpolated between, or enabling a search for a jointly optimized single "best metric." We will definitely investigate adding ImageReward to the final leaderboard given this.
> 2. Yes, this is another stimulating direction for future work. Using this human baseline also has implications for direct "VLM-as-a-judge" metrics that ask them to provide Likert scores, as only "superhuman Likert-assigning" VLMs would be sufficient to beat humble CLIPScore.
> 3. Agreed, we will make sure to clarify that QG/A metrics have this interpretability advantage, and in particular that this consideration, alongside TS2 evaluation, helps users choose a metric based on scenario and needs. Eg., in an interactive app (relatively low analysis throughput rate) cost is less of a consideration, and a human would benefit from interpretable analysis. Whereas for an online reward/feedback model or a supervisory post-filter for image generation, cost is an important consideration and interpretability isn't. Grounding metric selection in all of these considerations is best; TS2's contribution is in capturing one important consideration well.
>
> Thanks for the further stimulating discussion!

---

### Official Review · Reviewer_cvSS · 2024-07-14

**Soundness:** 3
**Presentation:** 3
**Contribution:** 3
**Rating:** 7
**Confidence:** 4

**Summary:**

The paper proposes an evaluation framework for holistically assessing text-to-image (T2I) evaluation methods. Since most of them are primarily established through simplistic correlational evidence and only compared to the CLIPScore baseline, this approach presents a more detailed way of assessment and also benchmarks existing promising evaluations. While there isn't a single clear winner, results suggest that CLIPScore is still a very competitive candidate and is especially successful when considering the much lower compute costs.

**Strengths:**

- important contribution to investigate and improve automatic T2I evaluation strategies
- interesting dataset construction which appears to build a more challenging test bed compared to previous methods, allowing more detailed insights and distinctions
- thoughtful discussion of the results & exploration of limitations
- very informative figure 2

**Weaknesses:**

- I like the general setup but I'm unsure about the accuracy of the "number of errors" counting system. I'll give two examples from within the paper. Take the last example in Figure 3 with the prompt "A gray elephant and a pink flamingo". An image with two flamingos is categorized as containing one error because there is no elephant. However, if there additionally was an elephant, it would still have an error since there are two instead of one flamingo. So one could argue that there are in fact two errors: a missing elephant and an additional flamingo. Or you say it's one error because there is one animal that is a flamingo but should be an elephant. So this is inherently ambiguous. However, in any ranking solution, this can actually matter quite a lot, so I'm worried that this introduces noise into the analysis process that is hard to reason over. I'm wondering to what extent this is taken into account by the design and how sensitive the results are to this. (Second example to illustrate from Figure 1 SEG: it's noted that when the shirt is not green, it's counted as one error. What if the shirt was additionally also suddenly a hoody. Is that then two errors or still only one? When the boy is gone overall, it's two errors because the shirt isn't green and there is no boy -- but what if there was now a grey shirt in the picture?)
- Given that the evaluation framework provides many different evaluations for varying setups and (as discussed in the paper) those might come with their own biases, what is the recommendation to those who are thinking about using this framework for when they can call their metric successful? Defining an overall evaluation aggregate might also help with the adoption of the framework.
- (Minor: It's hard to see which numbers are italicized in the results table.)

**Questions:**

None

**Limitations:**

Sufficiently addressed.

---

> ### Author Rebuttal · Authors · 2024-08-07
>
> Thank you for your detailed review! We appreciate your recognition of our *important contribution* which you find *interesting*, that enables *detailed insights* and has a *thoughtful discussion.* We hope you will find our response to your questions about the counting system satisfactory.
>
> ### Weaknesses
>
> - **Ambiguity introduced in the error counting approach**
>
> For count errors, we considered “a X” to remain correct if there is more than one “X”---a picture of multiple flamingoes does contain “a flamingo.” However, when specific numbers were provided in the prompt, like “one flamingo,” containing more than one counts as an error.
>
> You are right to point out that there is not necessarily a single objective answer for how to handle multiple attributes that could be incorrect about an object simultaneously, such as your example where “a boy in a green shirt” has a grey hoodie. In this case, under our annotation scheme, we still counted it as one error—”no green shirt,” but the case could be made for it to be two.
>
> While it is important to clearly document these annotation nuances for reproducibility, **these issues are inconsequential to our results, because our meta-metrics only evaluate rankings along descending walks in the error graph.** An image of the boy in a grey shirt and the image of a boy in a grey hoodie are not child or parent nodes of each other, instead being related only to images that may also have no boy at all. So regardless of whether a grey shirt or grey hoodie count as the same number of errors, a *grey shirt with no fruit* has more errors than *a green shirt with no fruit*, and has fewer errors than *no boy and no fruit* at all. The relative difference of errors between nodes that aren’t connected on the error graph do not actually matter, as our metrics are only assessed over walks.
>
> We will ensure this deeper documentation of the error counting process, as well as this explanation for the role the error counts play in evaluation (relative to directed connected nodes, absolute values unimportant) are both provided in the camera ready.
>
> - **What are our recommendations to future system builders**
>
> We have a few thoughts on this front. We think the right way to use TS2 in metric evaluation is to treat it as *an orthogonal evaluation axis to human preference.* While human preference correlations are great for capturing total output image quality, metrics that are well-correlated with TS2 will be ideal for measuring structural correctness. Authors might want to consider releasing dual metric evaluations, one which performs well on TS2, and another that performs well on image quality and aesthetic human preference correlations, to give more fine-grained feedback to models.
>
> The pareto optimality evaluation plays a particularly important role when introducing metrics that may be employed for automated feedback, such as in a reinforcement learning setup or as a post-filter to select the best candidate image from a set of outputs.
>
> As for metrics that may perform better on our evaluation, we think authors should consider better ways to gain faithful outputs from VQA modules, perhaps using techniques such as chain of thought or self-consistency prompting.
>
> Thank you for the stimulating question, we will include this discussion in our camera ready.
>
> - **Difficult to read italics in results table**
>
> Thank you for pointing this out. Rather than italicizing we will underline the runner-up scores.

---

> > ### Comment · Reviewer_cvSS · 2024-08-09
> >
> > I thank the authors for their response.
> >
> > What you're saying makes sense to me, especially when it comes to the error counting matter.
> >
> > I just want to reiterate on one part of my prior review which is on defining when this framework establishes "success". I understand that this framework provides a detailed holistic overview on a range of interesting dimensions (see Table 2). However, I'm wondering whether there is a recommendation for researchers who want to use this framework to choose the best-performing solution. Are there specific rankings/dimensions that are most diagnostic for overall performance? (And for adoption in the broader community, having an overall score that accumulates the individual results in Table 2 might help with adoption in the community. Do you have a suggestion what this score might be?)

---

> > > ### Author Response · Authors · 2024-08-09
> > >
> > > Thanks for a quick follow up! To give a more committal answer your question:
> > >
> > > Intuitively, the walk-based spearman correlation metric is probably the best choice for an overall score, as it captures the core desideratum of "able to correctly compare similar images by structural differences." For this desideratum, higher is always better.
> > >
> > > While the other two scores do matter---it is important that adjacent nodes be statistically significantly separated---it is less clear that higher is always better, vs anything over a threshold being sufficient. Thus a good recommendation might be to rely on the walk ordering score (which is also the main novel contribution here) and to treat the separation scores as a secondary consideration.
> > >
> > > In other words, par performance on the separation metrics alongside significant gains on ordering would be a very positive development, whereas significant improvement along separation with a loss in ordering could be negative---exact reverse ordering of the nodes with statistically significant node separation would get high delta scores, but be very bad.
> > >
> > > This is the recommendation and justification we will provide in the camera-ready: **a metric is clearly superior to others when it presents significantly higher ordering (particularly over the hard *nat* subset) without a significant drop in separation scores**---equivalent separation is sufficient.
> > >
> > > Ultimately, this recommendation is a judgement call and its main grounding is the aforementioned theoretical analysis (high ordering score always captures a good correlation to the scores, whereas high separation score can be present, even when the ordering is reversed). We will use this analysis to justify our recommendation to researchers in the conclusion section of the camera ready.

---

> > > > ### Comment · Reviewer_cvSS · 2024-08-13
> > > >
> > > > I thank the authors for their clarifications in the rebuttal. They sufficiently addressed my concerns and I updated my scores accordingly.

---

### Official Review · Reviewer_vRNq · 2024-07-21

**Soundness:** 2
**Presentation:** 3
**Contribution:** 3
**Rating:** 5
**Confidence:** 3

**Summary:**

The paper introduces T2IScoreScore (TS2), which aims to evaluate how good newly developed text-to-image (T2I) evaluation metrics/methods are. The authors formalize the task of evaluating t2i metrics as their abilities to *order* images correctly within SEGs.

**Strengths:**

1. The authors identify a very important task -- to evaluate T2I metrics. The introduction of T2IScoreScore and the use of semantic error graphs (SEGs) to evaluate T2I faithfulness metrics are novel and innovative.

2. Experiments are good. The methodology is rigorous, and the experiments are well-designed to test the core claims of the paper.

**Weaknesses:**

1. Limited Scope: The evaluation is primarily focused on a specific subset of T2I models (many variants of SD, and Dalle-2) and metrics. Expanding the scope to include a broader range of models and datasets would strengthen the generalizability of the findings. Potentially should consider other synthetic images from models such as OpenMUSE or aMUSEd (https://huggingface.co/blog/amused) with totally different generation architectures than diffusion, etc. Alternatively, text-2-image is an old task, even GAN and VAE can probably have image distribution other hand SD and Dalle-2 which heavily depends on CLIP.

2. Intrinsic Bias: The reliance on rank-correlation metrics, which have intrinsic biases, might affect the evaluation results. A more thorough discussion of these biases and potential alternatives could enhance the robustness of the conclusions.

**Questions:**

Instead of pairwise ranking/comparisons, which might not always be robust, have the authors design multi-images ranking instead of only two images for pairwise comparisons? Something like bradley-terry style ranking could make the evaluation more robust.

---

> ### Author Rebuttal · Authors · 2024-08-01
>
> Thank you for your thoughtful review. We are heartened by your recognition of our **rigorous methodology and well-designed experiments** approaching the important task of T2I metric assessment. We will briefly address your weaknesses and questions:
>
> ### Weaknesses
>
> - **Scope of image-generating models**
>
> Indeed there are many other generative image systems out there beyond the set we used to produce SEG-populating images. While it would be interesting to include images from OpenMUSE or aMUSEd in this approach, we believe showing a metric’s poor performance to rank on *any* set of T2I models is sufficient to demonstrate a need for improvement. Though we do not use a comprehensive set of T2I models to produce the test SEGs, we never claim our test is comprehensive, **only that it is sufficient in showing the surprising lack of differentiation between tested methods**.
>
> However, in the future images from additional SOTA models should be added to TS2 as advanced metrics are introduced. Our findings stand well with the current images but in future work we will definitely add more.
>
> - **Bias in the rank-order metrics**
>
> As we note in sec 6.1, Spearman’s rho (rank order corr) is indeed biased in favor of discrete scoring methods over continuous methods such as CLIPScore, because of the way they handle ties. Luckily, the continuous methods are the weak baseline that the QG/A and Caption-based methods (which are discrete) are intended to beat. The ranking score does indeed penalize these metrics---**yet they still compete with or outperform the rewarded metrics**. This makes the finding even more striking; despite being at a disadvantage, CLIPScore and ALIGNScore still win, to our surprise.
>
> In the future, as we maintain the TS2 leaderboard and other metric papers use it, it will become important to add metrics that do not penalize continuous metrics, as perhaps more advanced ones will be introduced. For the results in this paper though, **the bias actually strengthens the conclusion**.
>
> We will clarify this point in the camera ready.
>
> ### Questions
>
> - **Pairwise vs multi-image ranking**
>
> You ask about multi-image ranking instead of pairwise ranking. To clarify, *we do not do two image pairwise ranking*, **all of our evaluation metrics are multi-image-based**. Indeed, this is our work’s primary advantage over all prior T2I metric evaluations.
>
> Our metrics are either **full-walk** scoring ($rank_m$) or **node-pair** scores ($sep_m$ and $delta_m$). For the full-walk scores, ranking over each descending sequence of nodes (each of which contains multiple images)
>
> We only refer to pairwise comparisons in Table 1 as a way to distinguish our benchmark against prior work. Our metrics are exclusively multi-image (and our *per-equiv pref* score in Table 1 represents that on average, each node contains 3.4 images of equivalent correctness)
>
>
> ### Future updates to TS2
>
> We are excited to incorporate these points into the camera ready, and your suggestions for additional ranking metrics and more images are great ideas for future contributions as we expand TS2 as a living resource. However, we believe the resource as it currently stands represents a significant and timely contribution that advances our understanding of **current** text-to-image faithfulness metrics.

---

### Author Rebuttal · Authors · 2024-08-07

We appreciate all reviewers’ thoughtful and detailed analyses of our work.

We are excited that multiple reviewers identified each of our work’s key strengths, including:
1. That our meta-evaluation setting is a **timely and important task** (vRNq) that has not been approached before and is “often missed” (h3ZX) and constitutes an **important contribution** (cvSS) to the T2I evaluation field more broadly
2. That our approach is **novel and innovative** (vRNq), using an interesting dataset (cvSS) that is *quite valuable for future work* (h3XZ)
3. That our methodology is rigorous (vRNq) and “quite sound” (h3ZX)
4. That our “experiments are well-designed to test the core claims” (vRNq) and are comprehensive for including a wide set of VLMs in evaluating VLM-based metrics (Xy9h)
5. That our findings are **detailed** (cvSS), surprising and interesting, and our analysis is “very sound” (h3ZX)
6. That our discussion is thoughtful (cvSS) and our paper is “clearly written and easy to read” (h3ZX).

In particular we are pleased to hear that rev. h3ZX liked our paper overall and hopes to see it accepted!

Furthermore, **we are thankful for your thought-provoking and diverse questions and suggestions** implied in your weaknesses. We believe we have solid answers for most of your suggestions, and that the necessary changes to address them will greatly strengthen our paper. We hope that you find our answers useful or convincing.

Thank you!

---

### Decision · Program_Chairs · 2024-09-25

**Decision:**

Accept (spotlight)

**Comment:**

**Summary:** This paper addresses the issue of how to evaluate evaluation metrics for text-to-image models. They introduce a dataset of error graphs, which are prompts alongside images of with increasing number of semantic error graphs, such as incorrect color or count of objects. Evaluation metrics are scored by how accurately they rank images. Images that are more consistent with prompt, e.g. higher in the graph, should be ranked higher as well. They find that simpler, embedding space metrics such as CLIPScore tend to outperform more complex, GQ/A based metrics.

**Aggregate Strengths/Weaknesses:** The reviewers find that this is a structured approach to evaluating T2I metrics. They view the datasets as a valuable contribution and the experiments as in-depth and rigorous. The weaknesses are that there is not a single cohesive takeaway when comparing metrics using TS2. It’s not clear how researchers should decide on a single metric and TS2 also does not manage to incorporate other aspects of each metric, such as the fine-grained analysis GQ/A metrics provide. As a smaller point, the reviewers suggest including more models (even GANs/VAE) and more metrics (VQAScore, ImageReward). Time-permitting, it would be nice to see VQAScore in the final version. I'm especially curious about the results given its simplicity (no question generation, the difficulty of GQ/A approaches) while still leveraging a VQA model.

**Recommendation:** Every reviewer recommended this paper for acceptance. This work addresses an important problem of better understanding and selecting our evaluation metrics given the rise of T2I models. The dataset is well-structured and makes comparing two different metrics straightforward. The paper was well-received by all reviewers. I therefore recommend this paper for acceptance.